# Exploratory trajectory inference reveals convergent lineages for CD8 T cells in chronic LCMV infection

Jan T. Schleicher[1,2,3,4☯], Revant Gupta[1,2,3,4☯], Dario Cerletti[5,6☯], Ioana Sandu[5,6], Annette Oxenius[6‡*], Manfred Claassen[1,2,3,4‡*]

**1** Department of Internal Medicine I, University Hospital Tübingen, Tübingen, Germany, **2** M3 Research Center, University Hospital Tübingen, Tübingen, Germany, **3** Department of Computer Science, University of Tübingen, Tübingen, Germany, **4** Institute for Bioinformatics and Medical Informatics, University of Tübingen, Tübingen, Germany, **5** Institute of Molecular Systems Biology, ETH Zürich, Zürich, Switzerland, **6** Institute of Microbiology, ETH Zürich, Zürich, Switzerland

☯ These authors contributed equally to this work.
‡ AO and MC also contributed equally to this work.
* aoxenius@micro.biol.ethz.ch (AO); Manfred.Claassen@med.uni-tuebingen.de (MC)

**Data availability statement:** Raw sequencing data is available on Bioproject (accession number PRJEB43201). Processed data is available on Zenodo at https://doi.org/10.5281/ zenodo.10559456. Code for reproducing

## Abstract

Trajectory inference refers to the task of reconstructing state sequences of dynamic processes from single-cell RNA sequencing (scRNAseq) data. This task frequently results in ambiguous results due to the noisiness of the data. While this issue has been alleviated by the incorporation of directional information from RNA velocity analyses, it remains difficult to resolve complex differentiation topologies, such as convergent trajectories. We introduce exploratory trajectory inference to address this challenge. This approach considers unsupervised clustering analysis of trajectory ensembles derived from simulation-based trajectory inference to deduce differentiation lineages in a data-driven fashion. We assess this approach to resolve the convergent differentiation trajectories in CD8 T-cell differentiation in chronic infections. We utilize an original scRNAseq time-series dataset of CD8 T cells collected during the time course of a chronic LCMV infection. Simulation-based trajectory inference identified a branch region early during chronic infection where cells separate into an exhausted and a memory-like lineage. Exploratory trajectory inference further allowed us to identify a convergent differentiation trajectory traversing memory-like states and ending in the exhausted population. Adoptive transfer experiments showed CD8 T cells with predicted memory-like fate differentiating into both memory-like and exhaustion states, confirming the convergent differentiation topology. We expect exploratory trajectory inference to be applicable in other scRNAseq-based studies aiming at comprehensive characterization of differentiation trajectories with bifurcating and convergent topologies.

## Introduction

Biological processes such as cell type differentiation [1–3], immune response [4] or cell division [5] can be conceptualized as temporal sequences of coordinated, phenotypic state

the analyses is available at
https://github.com/claassenlab/Tex_repro.
Cytopath is available at
https://github.com/aron0093/cytopath and from
PyPI (https://pypi.org/project/cytopath/).
Cy2path is available at
https://github.com/aron0093/cy2path and from
PyPI (https://pypi.org/project/cy2path/).

**Funding:** This work was supported by the ETH
Zürich (grant no. 470 ETH-39 14-2 to M.C. and
A.O.) and the Novartis Foundation for
Biomedical Research, 471 DFG CL 792/1-1 and
the Center for Personalized Medicine (ZPM) and
DFG EXC 2180. The funders had no role in study
design, data collection and analysis, decision to
publish, or preparation of the manuscript.

**Competing interests:** No competing interests to
declare.

changes in the context of, possibly heterogeneous, cell populations. Such phenotypic states can be characterized by, e.g., epigenetic, transcriptional, and proteomic cell profiles. Furthermore, these differentiation processes are often asynchronously triggered. The differentiation processes give rise to state sequences with varying topologies including bifurcating, multi-furcating, cyclical, and convergent trajectories.

Single-cell approaches in conjunction with computational approaches have been used to study the state sequences of such processes. Single-cell RNA sequencing (scRNAseq) has increasingly become the experimental approach of choice due to its increasingly generic applicability. Different computational approaches have been proposed to model differentiation processes from scRNAseq data, specifically covering the tasks of pseudotime estimation, trajectory inference, or cell fate prediction. The goal of cell fate prediction is to determine the terminal differentiation state (fate) of any cell, possibly already early in the differentiation process. Such methods generate a score or probability per cell with respect to terminal differentiation states [6,7]. Pseudotime estimation addresses the task of ordering observed cells into a sequence of cell states traversed by a differentiation process. Typically, the estimated pseudotime values are interpreted as temporal ordering, not capturing the pace of differentiation. While pseudotime estimation might constitute sufficient characterization of a linear differentiation process, the description of complex processes with more involved topologies such as bifurcations requires an additional step of trajectory inference. Trajectory inference methods seek to infer a representative sequence of states that characterizes the possibly multiple differentiation processes in branching or convergent differentiation [8–11].

While contemporary trajectory inference approaches are based on static expression profiles, these profiles are ambiguous with respect to the directionality of potential cell state transitions. This ambiguity complicates data-driven assignment of root and terminal states without previous knowledge about the process, as well as resolving complex, i.e. cyclical [5] or convergent [3] process topologies. RNA velocity analysis enables estimation of transcriptional activity from scRNAseq data [1], consequently enabling the inference of likely transitions between different cell states in a data-driven fashion. These transcriptional activity estimates have been used to enhance the above reconstruction approaches, and in particular to improve pseudotime and cell fate prediction [6,12] and trajectory inference [4,13] and have been demonstrated to resolve aforementioned complex process topologies [4].

Despite these advances in trajectory inference approaches, resolving complex process topologies with slow and rare state transitions remains challenging. Here, we report exploratory trajectory inference to overcome this challenge. Specifically, we propose to use RNA velocity-derived Markov chain models to simulate and cluster ensembles of putative differentiation trajectories to identify differentiation lineages comprising slow and rare state transitions. We demonstrate this methodology to detect the replenishment of the T cell exhaustion compartment from memory-like CD8 T cells in chronic lymphocytic choriomeningitis virus (LCMV) infections.

## Results

### Exploratory trajectory inference overview

Exploratory trajectory inference is based on RNA velocity analysis of a scRNAseq dataset. Specifically, the resulting cell-to-cell transition probability matrix is used to define a Markov model of the studied differentiation process [1,14].

The objective of exploratory trajectory inference is (1) to sample an ensemble of cell state sequences from the Markov model, assuming as initial states the origin of the differentiation process under study, and (2) to summarize the ensemble to the level of major differentiation

lineages. Initial states are derived from a Markov random-walk model utilizing the transition probability matrix itself, as described in [1], or can be supplied by the user based on suitable prior knowledge. Sampling of cell state sequences is performed based on the transition matrix probabilities. The resulting cell state sequences are clustered using hierarchical clustering with dynamic time warping distance. The resulting clusters correspond to the tentative differentiation hierarchies (Fig 1).

## Landscape of CD8 T cells during chronic LCMV infection

We demonstrate exploratory trajectory inference for reconstructing CD8 T cell differentiation in chronic viral infection and in particular to assess the capability to detect the slow and

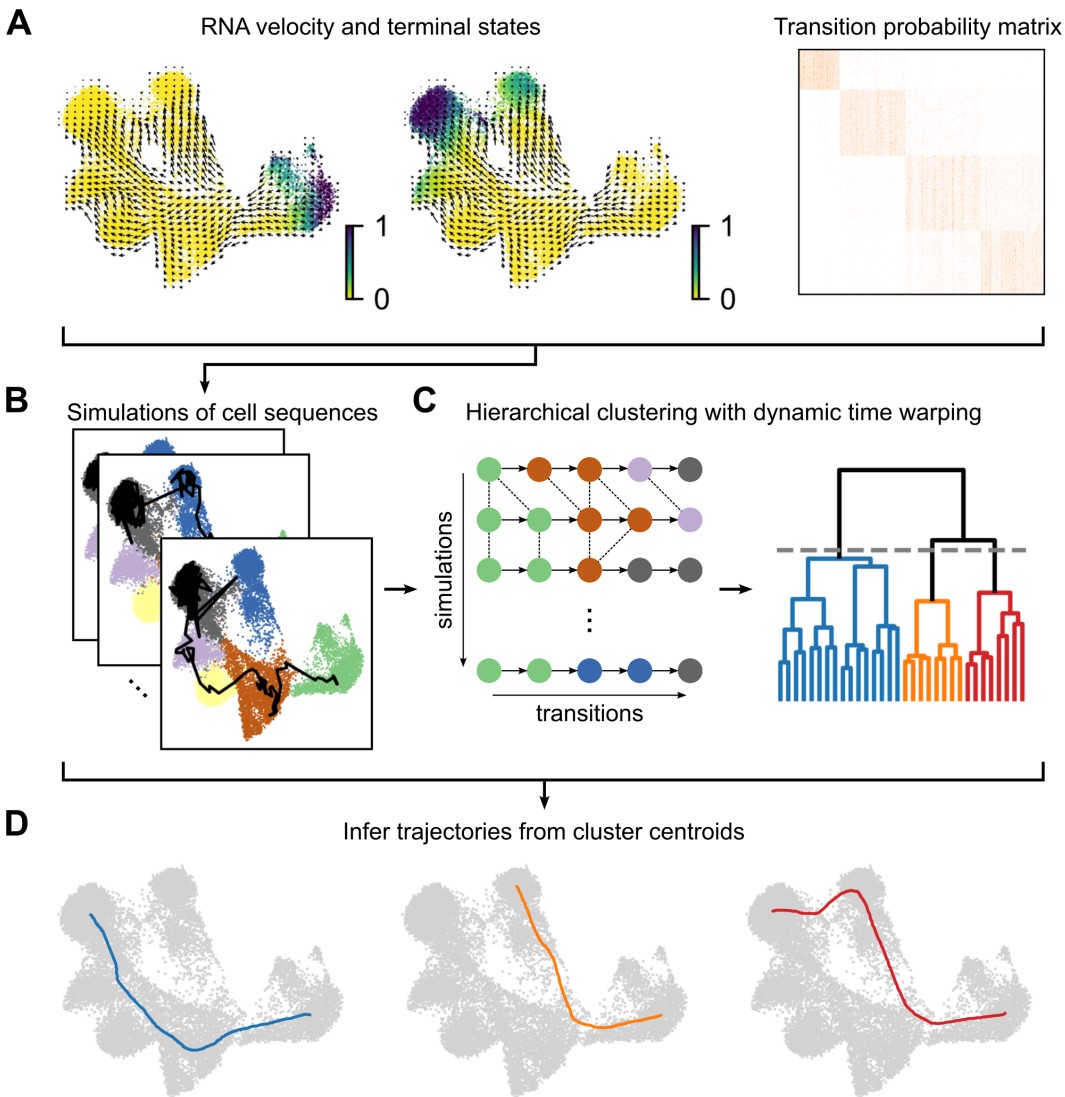

**Fig 1. Overview of exploratory trajectory inference.** (A) RNA velocity analysis, start- and end-point analysis are used to define a Markov chain model of the underlying cellular dynamic process. (B) The Markov chain model is used to simulate an ensemble of cell state sequences. (C) Cell state sequences are aligned and clustered via dynamic time warping. (D) Cluster centroids identify main cell state trajectories of the underlying dynamic process.

rare events of memory-like CD8 T cells replenishing the compartment of exhausted CD8 T cells.

Viral infections with human immunodeficiency virus (HIV), hepatitis C virus (HCV), and in mice with lymphocytic choriomeningitis virus (LCMV) can result in chronic infection with ongoing viral replication and high antigenic load for weeks or months. The continuous exposure to antigen drives CD8 T cells into a functionally distinct phenotype, collectively termed exhaustion [15]. This state is characterized by functional, transcriptional, and epigenetic changes that result in the expression of co-inhibitory receptors such as PD-1, LAG-3, TIM3, CD39, 2B4, CD160, and the transcription factor TOX, decreased secretion of cytokines like IL2, INFγ, and TNF as well as reduced proliferation and survival [15,16]. Acquisition of this exhausted phenotype is a continuous and gradual process driven by excessive T cell receptor (TCR) stimulation [17]. One sub-population of virus-specific T cells acquires a phenotype that shares properties with memory T cells from acute infection and has been linked to the expression and activity of T cell Factor 1 (TCF1) [18,19]. In contrast to terminally exhausted or effector T cells, these cells retain proliferative activity and have better survival in the infected host [20].

We acquired single-cell transcriptomic data from multiple time points during chronic infection, covering the very early phases (day 1-4), peak phase (day 7), contraction phase (day 14), and late phase (day 21) (Fig 2A), to capture an increased spectrum of the transcriptional landscape during the course of the infection that would allow a time-resolved analysis of single-cell heterogeneity and possibly more accurate inference of differentiation trajectories of virus-specific CD8 T cells. To this end, T cell receptor (TCR) transgenic (tg) LCMV gp33-41-specific CD8 T cells (P14Nur77 CTY labeled cells) were adoptively transferred into naïve C57BL/6 mice, followed by infection with LCMV clone 13 (Cl13). Activated and expanded P14 cells were isolated at the indicated time points and subjected to single-cell RNA sequencing (scRNAseq) analysis. The data was processed and visualized with UMAP [21] as described in Method section Pre-processing.

The UMAP projection roughly ordered cells according to their experimental time (Fig 2A). While UMAP embeddings tend to distort cell-cell relations and do not allow statements about global distances, this shows that cells are most similar within time points, but also share similarities across adjacent time points. Leiden clustering [22] defined 13 distinct clusters across all samples (Fig 2B). Based on previously described markers for the exhausted subsets, such as CD160, CX3CR1, and TCF1 [15,18,20], we further aggregated these clusters into six phenotypic groups (Fig 2C). Activated cells from day 1 to 4 post-infection (dpi) clustered at one peripheral region in the data, termed in the following **early group** (Fig 2C). These cells presented expression patterns of proliferation as well as of exhaustion, indicated by expression of *Cdk2*, *Mki67*, *Cdca8*, but also *Cd160* (Fig 3A, 3B, S1 Fig).

Conversely, two distinct populations from the latest time point at 21 dpi clustered at the other extremes of the spectrum. Of these *late* endpoints, one population showed high expression of a number of inhibitory receptors, including PD-1 (*Pdcd1*), CD39 (*Entpd1*), LAG-3 (*Lag3*), and CD160 (*Cd160*) (Fig 3A, 3B), indicating a terminally exhausted phenotype [16]. This **terminally exhausted group** (Fig 2C) was composed of clusters from 14 and 21 dpi (Fig 2D, S2 Fig), had the highest expression of co-inhibitory receptors, and additionally showed high expression of the transcription factor *Eomes*. The other endpoint population showed high expression of the transcription factor *Tcf7*, encoding TCF1, the memory-marker *Il7r* as well as *Slamf6* (Fig 3A, 3B), revealing this cluster as the previously described memory-like population [18]. This **memory-like group** (Fig 2C) was composed of two clusters from day 7, 14, and 21 (Fig 2D, S2 Fig).

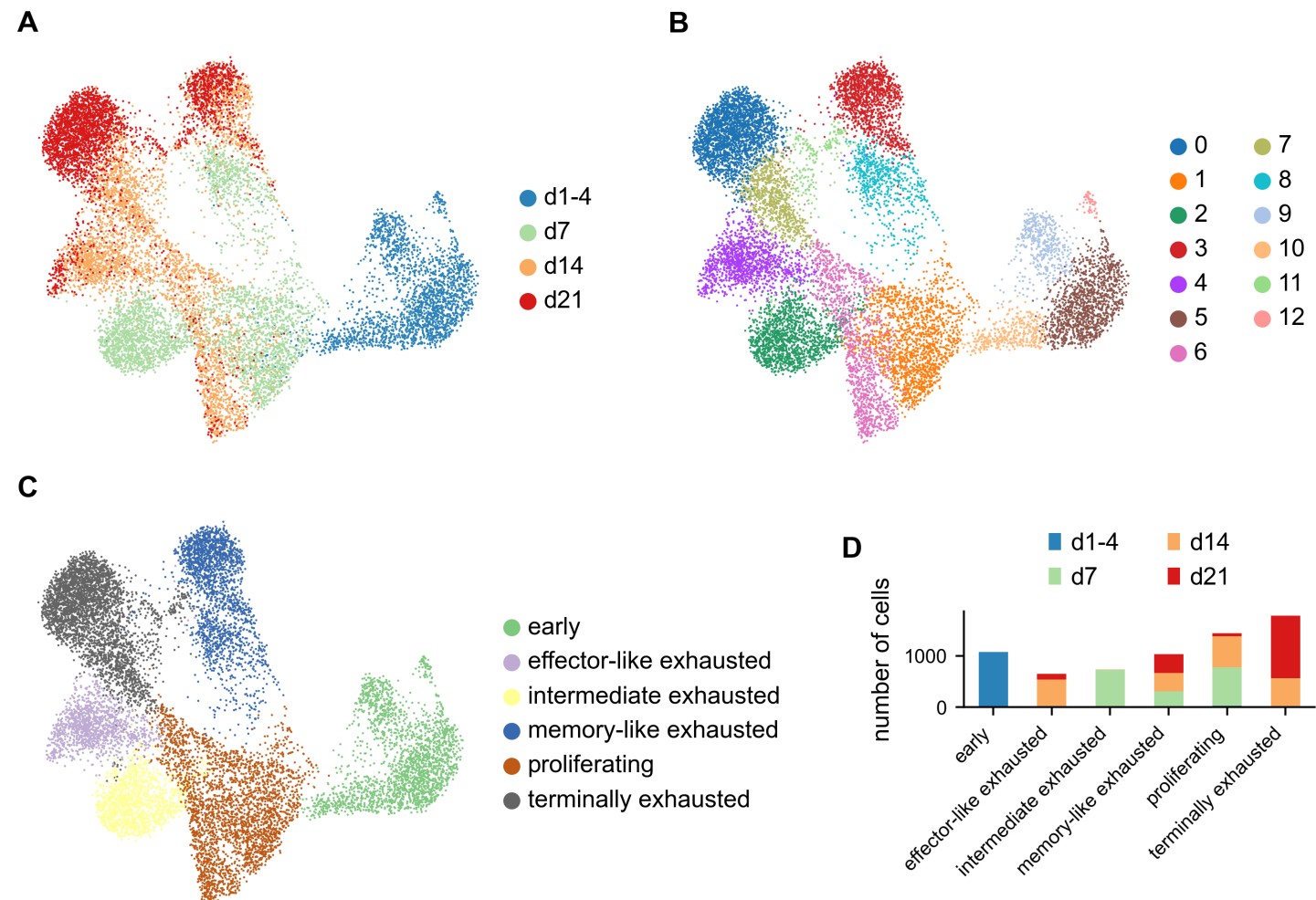

**Fig 2. scRNAseq time series of chronic LCMV infection.** (A–C) UMAP projections of scRNAseq data, colored by (A) the time point after infection at which cells were isolated for scRNAseq, (B) Leiden cluster assignment, and (C) phenotypic cluster annotation based on previously described marker genes and differentially expressed genes [20]. (D) Composition of the phenotypic clusters by sample time-point.

Cells from 7 and 14 dpi were situated between the 1-4 dpi and 21 dpi samples, with the 7 dpi samples being similar to cells from early time points on one end of the spectrum, but also connecting to splitting branches into exhausted and memory-like populations on dpi 14. At day 7, one cluster was identified that presented clear signatures of exhausted CD8 T cells but retained some expression of *Gzmb* as well as apoptotic genes like *Anxa1*. Hence, we termed this the **intermediate exhausted group** (Fig 2C).

At 14 dpi, some of the cells were still connected to the 7 dpi cell states, but a considerable fraction of cells had already further differentiated towards the two endpoints of 21 dpi. One cluster expressing *Cx3cr1* exclusively was termed **effector-like exhausted group** (cluster 4 in Fig 2B). This cluster showed elevated expression levels of *Cx3cr1* and of the killer lectin receptor genes *Klre1*, *Klra3*, and *Klrg1* (Fig 3A). These effector-like cells were only present in samples from days 14 and 21 (Fig 2D, S2 Fig).

Two clusters showed high expression of cell cycle genes such as *Mki67* (clusters 1 and 6) and formed the **proliferating group** (Fig 2C, Fig 3A). Further, we calculated scores for cell

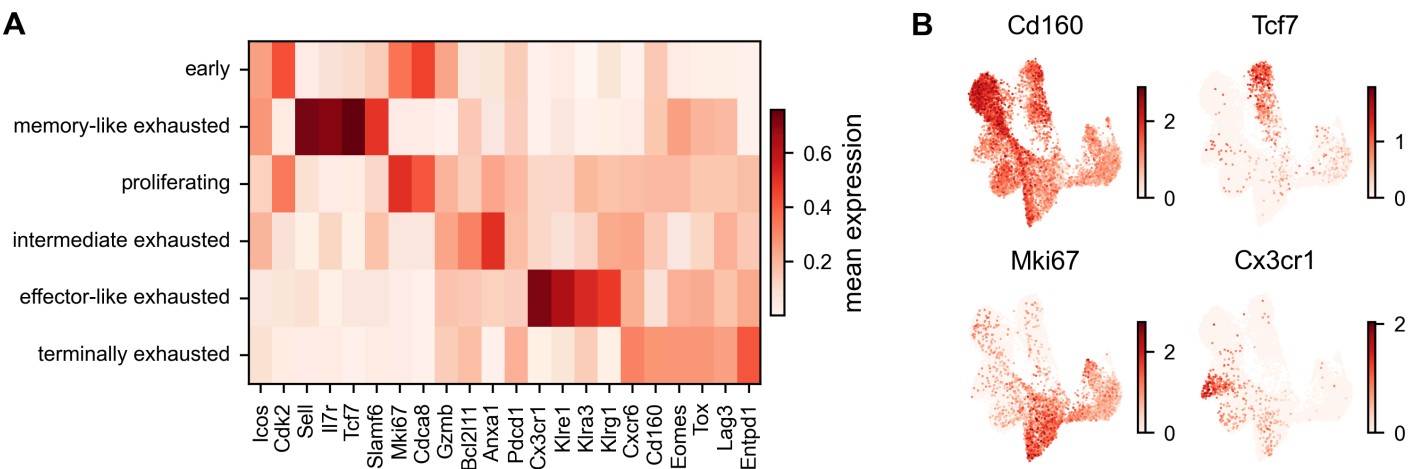

**Fig 3. Gene expression of characteristic markers.** (A) Heatmap of normalized gene expression for a selection of group-specific genes. The genes are grouped according to their phenotypic assignment. (B) UMAP projection of expression patterns of identified group-specific genes for the terminally exhausted (*Cd160*), memory-like (*Tcf7*), proliferating (*Mki67*), and effector-like group (*Cx3cr1*).

cycle and cell division genes based on the three different cell cycle stages G1, S, and G2/M (S3 Fig). The proliferating group presented high scores for S and G2/M phase and was composed of cells from 7 and 14 dpi and to a lesser degree from 21 dpi (Fig 2D).

Differentially expressed genes between phenotypic groups, calculated using Wilcoxon rank-sum tests, are shown in S4 Fig.

## RNA velocity analysis reveals developmental end points

Next, we aimed to assess the presence of developmental end points. To this end, we performed RNA velocity analysis [1]. We used scVelo to compute a transition probability matrix (TPM), demarcating likely transitions between cell states. For visualization and interpretation purposes, the TPM was summarized by summing up its columns and taking the mean of its rows to show transitions between phenotypic clusters as a graph (Fig 4A). This analysis indicated a state flow from early activated cells at 1-4 dpi towards early (dpi 7) and late exhausted cells (dpi 14 and 21).

Additionally, computing terminal states from the TPM (see Method section RNA velocity) allowed us to assign cells that are most likely at the start and at the end of the differentiation process. The highest probability of a start region was at the edge region of the early group (Fig 4B). The highest probability for endpoints was in regions from dpi 21 in the terminally exhausted group (0.56 on average in cluster, maximum 1.0). Additionally, there was a local maximum in endpoint probability in the memory-like group from dpi 21 (0.15 on average, maximum 0.63). Both the terminally exhausted cells as well as the memory-like cells seemed to comprise an endpoint of differentiation.

## Simulation-based inference reveals trajectories towards exhausted and memory-like phenotypes

The transition probability matrix based on the high-dimensional RNA velocity vector field was used for simulation-based inference of differentiation trajectories from cells in the start region [4]. This resulted in a temporal ordering of the cells and scores to which trajectory

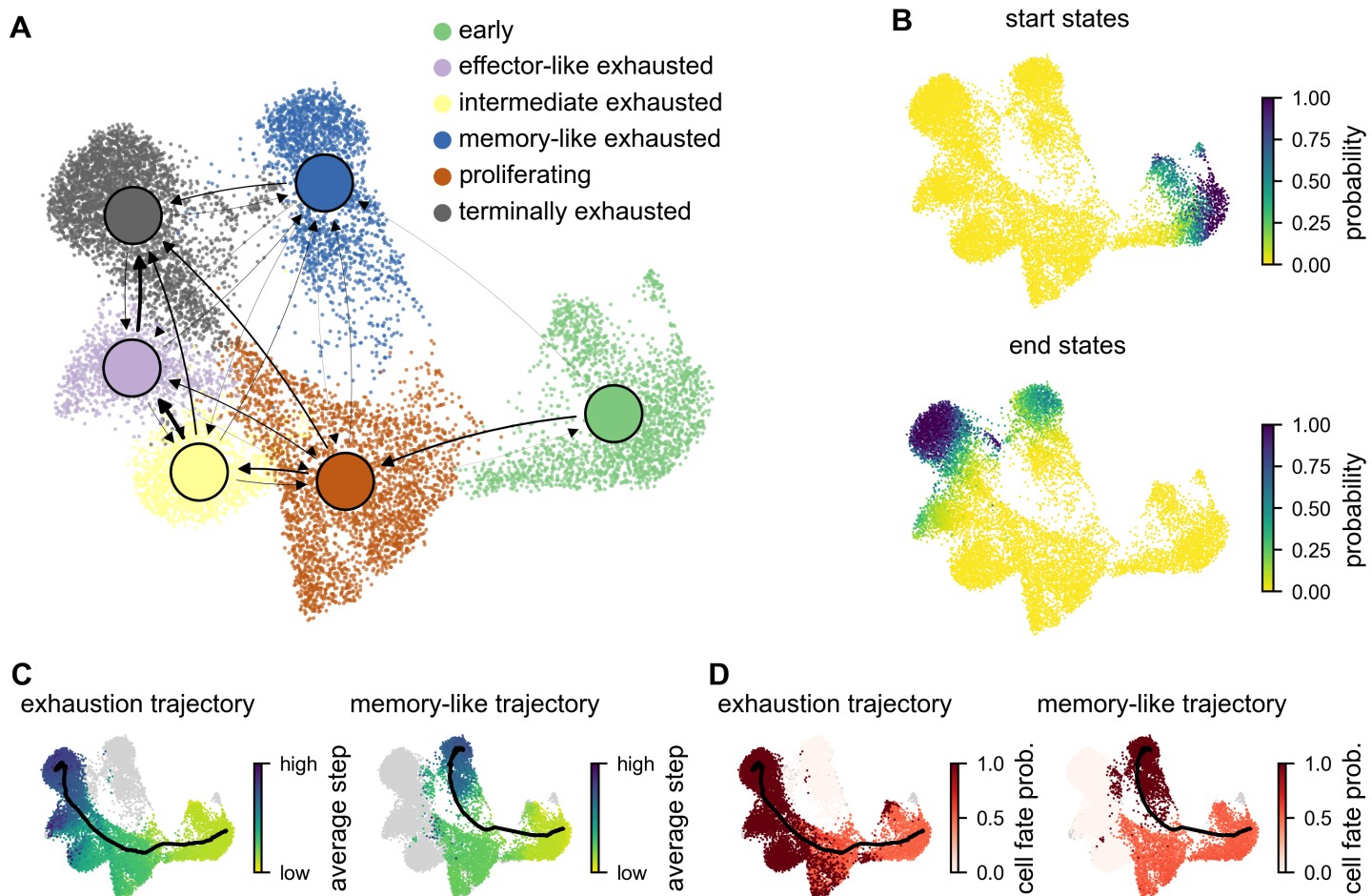

**Fig 4. Trajectory and cell fate inference.** (A) The RNA velocity-derived cell-wise transition probability matrix was summarized to yield transition probabilities between phenotypic clusters. A graph was constructed by drawing directed edges between clusters when the corresponding probability exceeded $10^{-3}$. The resulting graph with edge width proportional to transition probability shows a flow from early to memory-like and terminally exhausted cells. (B) Start and end states of the Markov process calculated from velocity-based transition probabilities. (C-D) Trajectories computed from simulations towards the two endpoint populations using Cytopath. Color represents (C) average simulation steps to arrive at a cell and (D) alignment-score-based cell fate probabilities. Average trajectory coordinates after alignment are depicted as black lines.

they belong (Fig 4C). From the alignment score, we computed the probability of a cell either acquiring memory-like or exhausted fate (Fig 4D).

Our analysis revealed two main trajectories, one towards the exhausted endpoint, and the other to the memory-like endpoint (Fig 4C, 4D). Both trajectories shared the same cell populations up to the proliferating group. From thereon the trajectories started to diverge into the two phenotypic branches. Multiple genes were found to be differentially expressed between the two endpoints (S5 Fig). In the memory-like group, *Slamf6*, *Ccr6*, *Tnfsf8*, and *Xcl1* were expressed at higher levels, many of them showing gradually increasing expression towards the most differentiated endpoint (*Slamf6*, *Ccr6*, *Tnfsf8*) (S6 Fig). *Xcl1* was high at the start of both trajectories, then decreased, and in the end increased again exclusively in the memory-like trajectory. The exhaustion branch showed increasing expression of *Cxcr6*, *Ccl5*, and *Nkg7* as the trajectory progressed toward the endpoint. In addition to these differentially expressed genes, we also observed gradually increasing levels of *Tcf7* along the memory-like trajectory.

(S6 Fig). For the exhaustion-linked genes *Pdcd1* and *Tox*, we only observed minor differences with slightly increased expression towards the terminally exhausted endpoint.

## Exploratory trajectory inference identifies convergent differentiation trajectories

It has been shown that the pool of exhausted CD8 T cells is continuously replenished from the pool of memory-like CD8 T cells [18]. To further investigate possible transitions between these endpoints, we performed exploratory trajectory inference. Specifically, we simulated cell state sequences from the Markov chain model established above. However, we did not enforce simulations to reach predefined endpoints. Instead, we simulated 1,000 cell state sequences with a predetermined, automatically chosen number of steps (Fig 5A). To infer lineages in terms of aggregated trajectories from individual cell sequences in an unbiased way, we grouped the simulations with hierarchical clustering, using dynamic time warping distance and Ward linkage. This approach yields a full clustering dendrogram (Fig 5B). Any number of lineages can be obtained by setting a corresponding threshold to cut the dendrogram.

Here, we chose different thresholds to obtain between two and six lineages (Fig 5C–5D, S7 Fig A–C). Cutting the dendrogram to obtain two lineages results in convergent lineages starting in the early group and ending in the terminally exhausted cluster (Fig 5C). However, they follow different paths: the first lineage, which we will refer to as the exhaustion lineage, passes through the proliferating, intermediate, and effector-like exhausted subgroups (S7 Fig D). In contrast, the other lineage, in the following termed memory-like lineage, diverges from the former in the proliferating cluster, enters the memory-like group, and finally transitions into the terminally exhausted cluster (S7 Fig E). Accordingly, cells at the start of the differentiation process have the potential to follow either lineage, while intermediate clusters are specific to their respective lineage. The end states in the terminally exhausted group are again shared between the two paths (S7 Fig F–G). Other trajectory inference methods such as Slingshot [8] and Monocle [9,23] did not identify converging trajectories (S8 Fig A–B). Additionally, we tested CellRank with the VelocityKernel [6]. While CellRank does not infer trajectories in the same sense as our method, it can compute fate probabilities for all cells in the dataset. In contrast to the findings of exploratory trajectory inference, CellRank computed high probabilities for the terminal exhaustion fate for almost all cells in the dataset (S8 Fig C). Only very few cells were assigned to the memory fate (S8 Fig D). While these results may hint at the possibility of transitions from memory-like to terminally exhausted cells, they are not as clear in this regard as the trajectories inferred by exploratory trajectory inference. As a fourth comparison, VIA was used to infer trajectories [24]. VIA produced many links between different clusters, convoluting the analysis (S8 Fig E). While its inferred pseudotime, which is lower in the memory-like branch than for the terminally exhausted cells, may indicate the transition described above, it yielded more terminal states, including one in the memory-like branch.

Interestingly, with three lineages, we already observed redundancy: The memory-like lineage was further split into two paths from the early through the memory-like into the terminally exhausted population (S7 Fig A). Furthermore, one of these two sub-lineages was supported by only 81 individual simulations. Combined with the stochasticity of individual Markov chain simulations, this led to more unstable averaging and possibly artifactual transition behavior between terminally exhausted and memory-like populations. This particular lineage was not further subdivided with four to six clusters. Similarly, further redundancy, including subdivisions of the exhaustion lineage, arose with four, five, or six lineages (S7 Fig B–C, Fig 5B–5D), respectively.

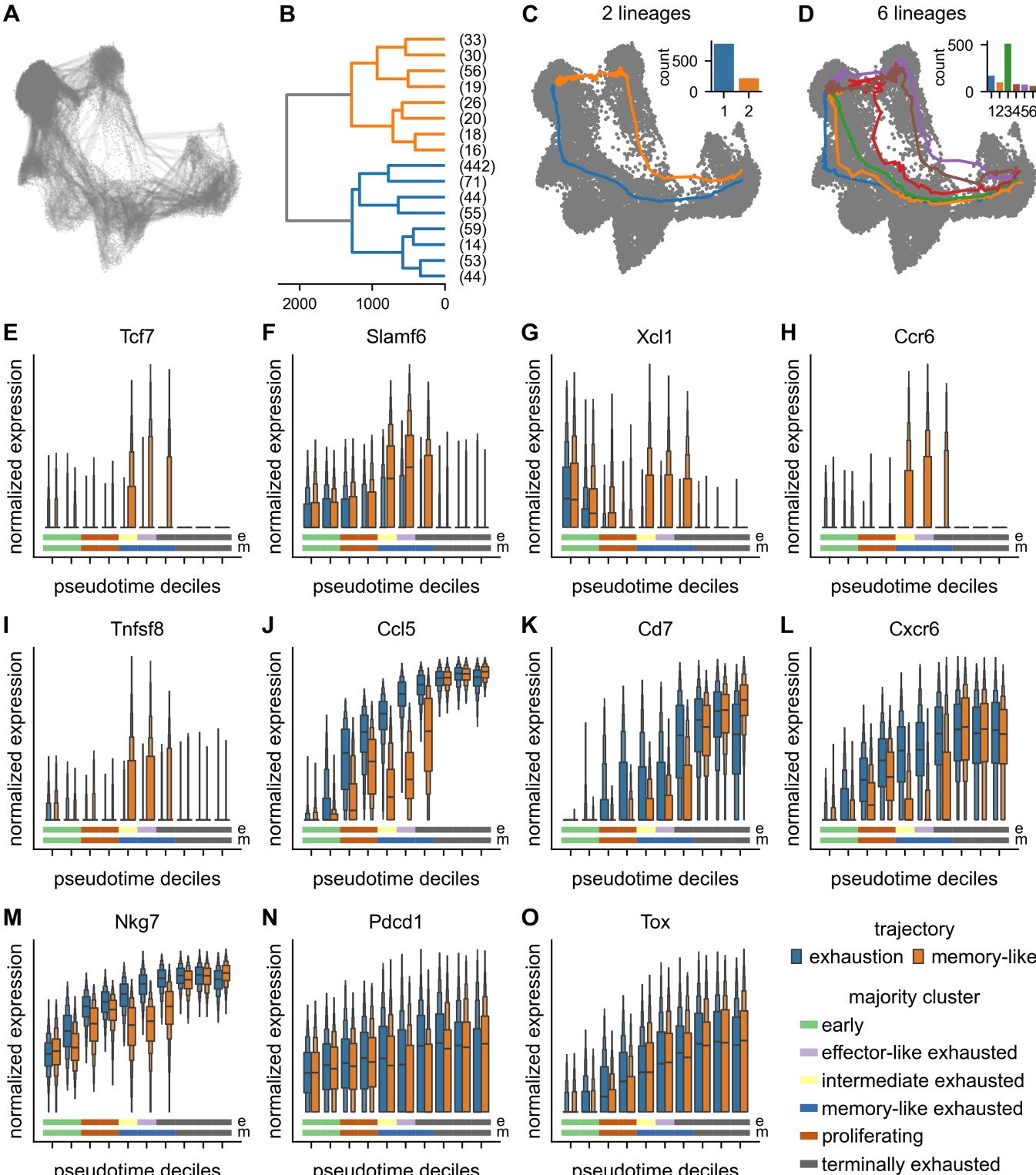

**Fig 5. Exploratory trajectory inference.** (A) Markov chains of cells sampled without enforcing fixed endpoints. (B) Individual cell sequences grouped with hierarchical clustering to obtain lineages. (C) Two distinct convergent trajectories toward the terminally exhausted group with cutoff set to yield two lineages. (D) Redundant, convergent trajectories toward the terminally exhausted group with cutoff set to yield six lineages. Barplots indicate number of cell sequences contributing to the respective lineage. (E-O) Expression of genes with differential expression patterns along exhausted and memory-like trajectories. Colored lines at the bottom indicate the majority cluster for each pseudotime decile for the memory-like (bottom, "m") and exhausted trajectory (top, "e"), respectively.

To assess the stability of inferred trajectories, we repeated the Markov chain simulations 10 times. In particular, in the case of two lineages, the results were stable, with a low deviation of inferred trajectory coordinates around the mean across all runs (S9 Fig). We found increasing uncertainty and overlap between lineages with an increasing number of lineages. The same trend was observed in PCA space in the form of the variance for each step across 10 runs, summed over all 50 principal components (S10 Fig). Additionally, we computed silhouette coefficients for the different numbers of lineages (S11 Fig). Together with the increasing uncertainty, these results indicate that the best number of lineages (with the highest silhouette coefficient) for this dataset is two.

As before, we compared gene expression along the trajectories for genes with differential expression between the two endpoints (Fig 5E–5O). Additionally, we also determined the majority cluster for each decile for the two trajectories. This analysis suggests that the exhaustion lineage transitions faster from the early to the proliferating cluster than the memory-like lineage and also reaches the endpoint slightly earlier. *Tcf7*, *Slamf6*, *Xcl1*, and *Ccr6* (Fig 5E–5H), differentially expressed in the memory-like cluster, showed an upregulation in the intermediate deciles of the memory-like lineage. Towards the end, when the trajectory transitioned into the terminally exhausted cluster, their expression levels decreased again to similar levels as in the exhaustion lineage. Genes that were upregulated in the terminally exhausted group such as *Ccl5*, *Cd7*, *Cxcr6*, and *Nkg7* (Fig 5J–5M), tended to increase in expression along both trajectories. However, their expression started to rise earlier in the exhausted trajectory and only converged in the last three pseudotime deciles, when the memory-like trajectory transitioned into the terminally exhausted cluster. *Pdcd1* and *Tox* only showed minor differences between the two lineages (Fig 5N–5O).

Finally, to assess the run time and memory requirements of the proposed exploratory trajectory inference approach across a larger range of dataset sizes, we resorted to a dataset of mouse gastrulation [25], downloaded from scVelo (version 0.3.3) [14]. To evaluate the relation between the number of cells and computational efficiency, the dataset was subsampled to different sizes (5,000, 10,000, 20,000, and 50,000 cells), and exploratory trajectory inference was run on each subsample dataset with four different settings for the number of chains (100, 500, 1,000, and 2,000 chains, S12 Fig). Due to the distance calculations between simulated chains, our method scales quadratically with the number of chains (S12 Fig D). The effect of the number of cells on run time (S12 Fig C) is linked to the number of steps required to reach the stationary distribution (S12 Fig E-F): With increasing size of the dataset, the number of required steps also rises.

In summary, exploratory trajectory inference indicates that the development of terminally exhausted CD8 T cells from cells putatively committed to memory-like states occurs via an intermediate transition through a memory-like state. Further, it allowed reconstructing transcriptional changes along this transition.

## Adoptive transfer experiments confirm transitions from memory-like to terminally exhausted cells

To experimentally corroborate the persistence of the developmental end points as indicated by RNA velocity analysis and the transition events across these, we set out to identify and later sort LCMV-specific CD8 T cells with phenotypes indicative of a pre-committed state or a committed state towards the exhausted endpoint or the memory-like endpoint.

Specifically, P14 cells were analyzed according to branch-specific markers CXCR6 and TCF1 for sorting the cells into pre-committed (CXCR6$^-$ TCF1$^-$), memory-like (CXCR6$^-$ TCF1$^+$) and exhausted (CXCR6$^+$ TCF1$^-$) cells (Fig 6A–6B). These two markers were chosen

based on their differential expression in the two presumed differentiation endpoints (Fig 3, S5 Fig) as well as on previous work describing TCF1 as a marker for the memory-like population [18]. We transferred naïve P14 T cells expressing GFP under the control of the TCF1 promoter into naïve C57BL/6 mice and infected the mice with high dose LCMV Clone-13 (Fig 6A). At 5 dpi, when all three populations of interest had formed, we sorted P14 cells from the three branches according to expression of CXCR6 and TCF1 (detected by GFP) and transferred them into infection-matched hosts. One week after transfer (12 dpi from the initial infection), we analyzed the progeny of cells originating from the three selected populations in the spleen (Fig 6C). We observed that cells recovered after transfer of early committed exhausted cells into infection-matched recipients retained their exhausted phenotype. Cells recovered after transfer of the early committed memory-like branch exhibited phenotypes of both exhausted and memory-like cells, confirming previous results of differentiation from memory-like into exhausted cells upon antigen encounter [20], and indicated partial plasticity among the cells with putative commitment towards the memory-like endpoint. Recovered cells after transfer of pre-committed cells exhibited both a memory-like or an exhausted phenotype, confirming their differentiation potential into both memory-like and exhausted cells. However, there was a strong bias towards differentiation along the exhaustion branch,

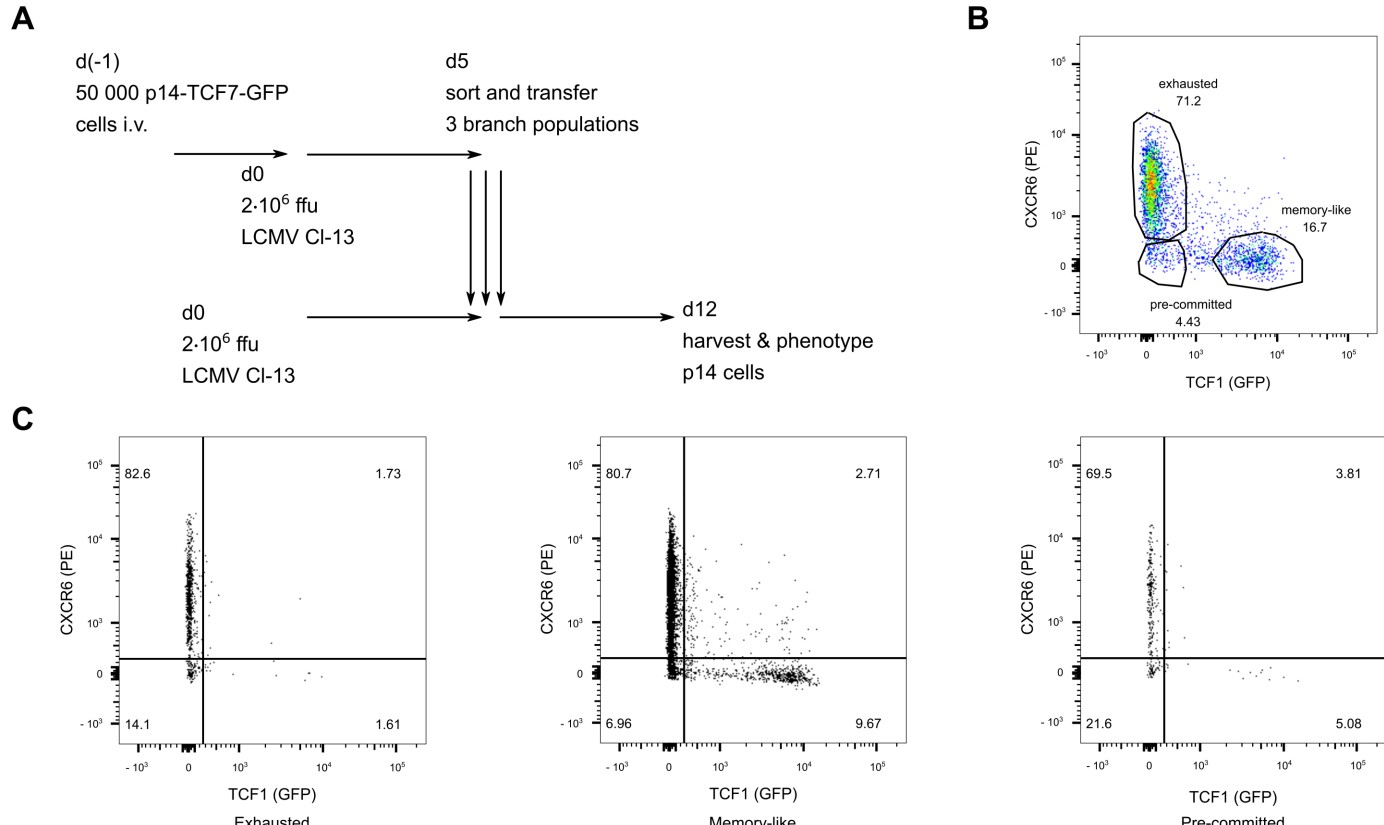

**Fig 6. Experimental analysis of early cell fate commitment.** (A) Naïve P14 cells were first transferred into naïve C57BL6 mice, followed by Clone-13 infection. At 5 dpi, the three P14 branch populations pre-committed (CXCR6⁻ TCF1⁻), memory-like (CXCR6⁻ TCF1⁺), and exhausted (CXCR6⁺ TCF1⁻) were isolated and adoptively transferred into infection-matched hosts. Their phenotype was assessed 7 days post transfer. (B) Flow cytometry gates used to sort exhausted, memory-like, and pre-committed branches. (C) Phenotype of the recovered P14 cells at 12 dpi from spleens after high-dose Clone-13 infection and transfer of either exhausted, memory-like, or pre-committed cell populations isolated at 5 dpi. Cells were gated on P14 cells.

which might be explained by the much more extensive proliferation of these cells compared to memory-like cells.

The experiments are consistent with the afore-postulated convergent trajectories, where the memory-like and terminally exhausted clusters are visited in the same trajectory.

## Discussion

Here, we describe exploratory trajectory inference for the reconstruction of complex and subtle differentiation trajectories. This approach extends simulation-based trajectory inference from RNA velocity fields [4] by considering the unsupervised analysis of state sequence simulations without predefined end points. This enables the data-driven definition of lineages that are possibly difficult to anticipate on the basis of predefined endpoints.

Exploratory trajectory inference does not require prior knowledge of cell types or state labels. The only prerequisite for performing simulations is the presence of a transition probability matrix and root and end state probabilities. In the absence of these, they are computed on the spot based on RNA velocity estimates, if available. For downstream analysis including alignment of cells to trajectories to infer pseudotime and cell fate, cluster information is required. Cell type annotations can aid the biological interpretation of inferred trajectories.

We use UMAP embeddings to visualize the results of exploratory inference. While changes in hyperparameters influence these embeddings, the results of exploratory trajectory inference are not affected by these parameters, since Markov chain simulations and their summarization are performed in PCA space.

We assessed exploratory trajectory inference for analysis of differentiation trajectories of virus-specific CD8 T cells during chronic LCMV infection using scRNAseq time-series data from seven time points covering activation, peak, contraction, and late phase of the response. The time-resolved transcriptional landscape revealed a proliferating population and cells with an intermediate and effector-like exhausted phenotype as transitional states of this process, respectively. The RNA velocity analysis allowed us to estimate transitions between the cell states during the progressing immune response. At the early time points until day seven, the two trajectories were nearly indistinguishable but then diverged increasingly toward their respective endpoints. These results complement more detail about the cell state sequences from trajectories reported in previous studies. Specifically, these studies consistently described the memory-like and the terminally exhausted cell state, which we also identified. Some studies [26,27] additionally described an effector-like CX3CR1+ population arising late during the infection also present in our late samples from 14 and 21 dpi.

The replenishment of exhausted CD8 T cells from the pool of memory-like CD8 T cells has been reported [18]. Two studies investigated plasticity and differentiation of the memory-like cells computationally by lineage inference on a single time point and through adoptive transfer experiments, concluding that memory-like cells partially maintain their phenotype and can give rise to terminally exhausted and effector-like cells [20,26]. While our initial lineage inference across multiple time points suggests that the exhaustion and the memory-like lineages are separate, exploratory trajectory inference supports the previous observation that the development of terminally exhausted CD8 T cells from CD8 T cells is also implemented via an intermediate transition through a memory-like state. This analysis further allowed the reconstruction of the detailed putative transcriptional changes along this transition. As for Cytopath, this convergent behavior was not identified by popular trajectory inference methods like Monocle and Slingshot [4,28], while CellRank [6] and VIA [24] hinted at this behavior, but yielded less conclusive results than exploratory trajectory inference.

We confirmed this convergent behavior with adoptive transfer experiments on the basis of markers that separate cells into the pre-committed (TCF1$^-$ CXCR6$^-$), exhausted (TCF1$^-$ CXCR6$^+$) and memory-like branch (TCF1$^+$ CXCR6$^-$). Adoptive transfer of cells sorted according to these markers, likely belonging to the three branches, at day five post chronic LCMV infection into infection-matched new hosts, confirmed the plasticity of pre-committed cells to acquire both exhausted as well as memory-like phenotypes. Conversely, progeny from committed exhausted cells largely retained their phenotypes. The adoptive transfer experiments indicated that CD8 T cells putatively committed to a memory-like fate partly develop into a terminally exhausted state, confirming the reconstructions from exploratory trajectory inference from the scRNAseq time series data.

Computational approaches such as the proposed approach require users to set parameters. We find that exploratory trajectory inference is robust over a wide range of parameter choices. For instance, RNA velocity inference is sensitive to changes in the parameters of the chosen inference method. We assessed the influence of different velocity estimation procedures on exploratory trajectory inference. For the three different models in scVelo ("deterministic", "stochastic", and "dynamical") and four different settings for the number of neighbors (15, 30, 50, and 100), the inferred velocities, root cells (S13 Fig), and end points (S14 Fig) were very similar, as confirmed by high pairwise Spearman's correlation (S15 Fig), and mostly differed in the position of the root cells within the early cluster and in the magnitude of end cell probabilities of the terminally exhausted and memory-like clusters. Further, the lineages inferred with exploratory trajectory inference (S16 Fig) were also stable across the different parameter settings. Notably, the only qualitative difference between the trajectories inferred from stochastic velocities across four different neighborhood sizes was the failure to detect the transition from memory-like to terminally exhausted cells with 15 neighbors. The dynamical model resulted in larger deviations for 15 and 30 neighbors, but produced similar qualitative trajectories for 100 neighbors. However, for the two strongly deviating results based on the dynamical model, the inferred trajectories also showed a higher variance across repeated runs than those for the stochastic model. Our findings reported in this study, obtained with the stochastic model and 30 nearest neighbors, were supported by the majority of trajectory inference analyses with the other considered parameters.

Another parameter that potentially influences the inferred trajectories is the number of steps in the Markov chain simulations. In general, this is chosen automatically based on the number of steps required to converge to the steady-state distribution, up to a tolerance threshold. Nevertheless, we systematically assessed the effect of chain lengths deviating from this choice on the inferred trajectories (S17 Fig). While a number of steps considerably lower than the convergence threshold (here: 228 steps) resulted in trajectories that did not reach the end points or did not show a transition from memory-like to terminally exhausted cells, longer chains still produced the same overall trajectories. We conclude that a chain length that is too low hinders correct lineage inference in the sense that terminal states are not reached by the simulations, but overly long chains do not strongly affect the inferred trajectories. Hence, we suggest setting a low tolerance threshold for convergence to ensure sufficiently long simulations.

Furthermore, we assessed the added value of a velocity-based transition probability matrix by using a matrix based only on neighborhood connectivities, computed with CellRank [6]. The analysis based on this matrix resulted in root cells spanning all early cells and end states spread out across the whole dataset (S18 Fig A-B). At the default threshold, the automatic selection of the number of steps did not converge, and the inferred trajectories were meaningless (S18 Fig C), demonstrating the importance of RNA velocity for exploratory trajectory inference.

While we demonstrate our approach for the detection of transitions of T cells from the memory-like to a terminally exhausted state, we further report the capability of exploratory trajectory inference to infer trajectories with diverse topologies with known ground truth. Specifically, we simulated data from processes constructed in 2D (see Simulation of positive control data for details on data generation). Exploratory trajectory inference faithfully reconstructed bifurcating, branching, converging, and even cyclical trajectories (S19 Fig). Moreover, automatic selection of the number of lineages through silhouette coefficients always yielded the correct number of ground-truth lineages. This approach constitutes a very well-defined, simplified baseline to assess the performance of exploratory trajectory inference. While showcasing the capability of our method to infer lineages in situations with velocities strongly aligned with the defined dynamic process, it does not represent the statistical properties of sparse scRNAseq count data. Therefore, to generate more realistic simulations with known trajectories, we additionally used dyngen [29], with various backbones, resulting in diverse process topologies. Here, exploratory trajectory inference showed overall good performance in recovering the true trajectories as well (S20 Fig). Automatic selection yielded the correct number of lineages for all backbones except for the cycle. Here, we obtained three lineages instead of one, all following the cyclic dynamic process. However, this simulated dataset was particularly challenging since the inferred end points are distributed all along the cyclic path, and the ground truth simulation also showed multiple cycles (S20 Fig D–F), in contrast to, e.g., the bifurcating backbone, where end points were more restricted and where there were no cyclical patterns (S20 Fig A–C).

As a negative control, we performed exploratory trajectory inference on a dataset of peripheral blood mononuclear cells (PBMCs), comprising a variety of immune cells that do not differentiate into each other [30]. Accordingly, we do not expect trajectories between them, and trajectory inference in this setting should not report such lineages. For this dataset, root cells and end points based on RNA velocity were very diffuse (S21 Fig A-B). Furthermore, the iterative procedure used to determine the number of simulation steps did not converge at the default, and not even at a 10 times higher threshold. These findings, which we also described above for other settings where exploratory trajectory inference failed, indicate the unsuitability of the data for trajectory inference. While exploratory trajectory inference can formally be executed in this situation and then still reports trajectories, these trajectories should not be considered for further investigation due to the aforementioned inherent unsuitability of this dataset for velocity-based trajectory inference (S21 Fig). Therefore, we suggest that potential users employ the described indicators of diffuseness of root/end states and non-convergence to judge whether to proceed with trajectory inference. To judge the diffuseness of terminal states, we suggest assessing the joint and marginal distributions of root cell and end point probabilities (S22 Fig). In a situation of low diffuseness of terminal states, the majority of cells has a root cell and end point probability of zero, few cells have a high probability for either state, the marginal distributions have a low median absolute deviation (MAD), and, most importantly, there are no cells with non-negligible probability for both terminal states. The T cell exhaustion dataset analyzed in this study exhibits the above properties indicative of a situation of low diffuseness of terminal states (S22 Fig A). For the PBMC dataset, we find a situation of high diffuseness, i.e., the majority of cells could, to a varying extent, be either a starting or an end state, and the marginal distributions have a higher MAD (S22 Fig B). The simulated dyngen cycle dataset lies between these two extremes, with well-defined root cells, but more widely spread end point probabilities (S22 Fig C). In our experience, these factors are useful determinants of whether exploratory trajectory inference is applicable for a dataset.

This work shows how scRNAseq time series data can be utilized to achieve a detailed reconstruction of differentiation processes with non-trivial topologies. Here, we focus on elucidating the development of T cell exhaustion in chronic LCMV infections. We present an unsupervised simulation-based approach to identify unusual differentiation trajectories. We demonstrate this approach to define a path for the replenishment of the pool of exhausted CD8 T cells from memory-like cell states in addition to the more direct differentiation of early activated CD8 T cells. Exploratory trajectory analysis is not confined to this process and is instead applicable for any study monitoring differentiation processes via scRNAseq and is expected to help discover non-trivial, possibly unexpected trajectories. This broader applicability was also demonstrated on simulated data with well-defined ground-truth trajectories of varying topologies.

## Materials and methods

### Infections and cell isolation

**Mice.** Wild-type male C57BL/6J mice were purchased from Janvier Elevage. Nr4a1-GFP mice expressing GFP under the control of the NUR77 promoter [31], P14 transgenic (CD45.1) mice expressing a TCR specific for LCMV peptide gp33–41 [32] were housed and bred under specific pathogen–free conditions at the ETH Phenomics Center. P14-Tcf7-eGFP mice were used for validation experiments to sort TCF1+ cells. All mice used in experiments were between 6–16 weeks old. P14-Nr4a1-GFP mice were generated by crossing Nr4a1-GFP and P14 mice. All animal experiments were conducted according to Swiss federal regulations and approved by the Cantonal Veterinary Office of Zürich (Animal experimentation permissions 147/2014, 115/2017).

**Virus.** LCMV clone 13 [33] was propagated on baby hamster kidney 21 cells. Viral titers of virus stocks were determined as described previously [34].

**Infection.** $10^4$ TCR transgenic cells (P14 or P14-Nr4a1-GFP) were adoptively transferred 1 day prior LCMV clone 13 intravenous (IV) infection with $2 \cdot 10^6$ ffu/mouse. For isolation of activated P14 cells at 1-4 days post-infection, $10^5$ CTY labeled P14-Nr4a1-GFP cells (CellTrace$^{TM}$ Yellow Cell Proliferation kit (Thermo Fisher) according to the manufacturer's protocol) were transferred and GFP+ (reporting TCR triggering) cells were isolated.

**Cell isolation from tissues.** After 1, 2, 3, 4, 7, 14, and 21 days of chronic infection, mice were sacrificed with carbon dioxide, and spleens were isolated. Spleens were mashed through 70 μm filters with a syringe (1 mL) plunger. Cell suspensions were filtered (70 μm) and treated with ammonium-chloride-potassium buffer (150 mM NH4Cl, 10 nM KHCO3, 0.1 mM EDTA in water) to lyse erythrocytes for 5 min at room temperature.

**Cell sorting.** Spleen samples were depleted of CD4 T cells and B cells by incubating splenocyte suspensions in enrichment buffer (PBS, 1 % FCS, 2 mM EDTA) with biotinylated α-CD4 (Biolegend, #100404 clone HK1.5) and α-B220 (Biolegend, #103204 clone RA3-6B2) antibodies at room temperature for 20 min, followed by incubation with streptavidin-conjugated beads (Mojo, Biolegend) (4%) for 5 min at room temperature. Cells were then placed on a magnetic separator (StemCell) for 10 min at room temperature, followed by collection of supernatant. For scRNAseq, cell suspensions of spleens from five mice were pooled in samples from day 7, 14, and 21 post-infection and from three mice for samples from day 1, 2, 3, and 4 to ensure the samples were representative. All samples from days 1 to 4 were pooled for sorting and sequencing due to the low frequency of P14 cells. Enriched samples from the spleen were stained with α-CD8-PerCP, α-CD45.1-APC and fixable LiveDead (Thermo Fisher) dye to sort live P14 cells (ARIA cell sorter, BD Biosciences).

## Single-cell RNA sequencing and analysis

Sorted P14 cells from different time points were washed and resuspended in 0.04% BSA. Single-cell sequencing was performed at the Functional Genomics Center Zurich. Cell lysis and RNA capture were performed according to the 10X Genomics protocol (Single Cell 3' v2 chemistry). The cDNA libraries were generated according to the manufacturer's protocol (Illumina) and sequenced (paired-end) with NovaSeq technology (Illumina). Transcripts were mapped with CellRanger (10X Genomics; version 2.0.2).

**Pre-processing.** Reads were realigned and spliced and unspliced counts were determined with velocyto [1]. Scanpy [35] was used to analyze scRNAseq data. Genes with less than 20 shared counts (spliced and unspliced) were removed. Cells were filtered to have a minimum of 500 and a maximum of 3,500 genes, and at most 15,000 total counts. Additionally, cells with more than 2% mitochondrial counts, less than 10%, or more than 40% ribosomal counts were removed. The gene expression matrix was normalized, log-transformed, and scaled as described in [30]. Cell cycle scores were determined based on 97 genes [36]. To remove the effect of the cell cycle on gene expression, the difference between S and G2/M scores was regressed out. The 5,000 most variable genes were determined in Scanpy with the CellRanger method [30]. PCA was performed based on these highly variable genes, and a neighborhood graph with 30 nearest neighbors was constructed based on 50 principal components. Leiden clustering and UMAP were computed using standard parameters.

**RNA velocity.** ScVelo [14] (version 0.2.5) was used to estimate RNA velocity and infer cell transition probabilities. We estimated gene moments using neighborhood connectivities from 50 principal components, and 30 neighbors with the UMAP method. Velocity was inferred using "stochastic" mode. A velocity graph was computed using nearest-neighbor connectivities. Terminal states and a transition probability matrix between cells were computed using scVelo, without self-transitions.

**Cytopath.** We applied scVelo's terminal states routine to compute equilibrium distributions of the forward and backward Markov process, excluding self-transitions. Regions with high terminal state probability were identified and the corresponding Leiden clusters were used as start and end points for trajectory inference. Specifically, we used Cytopath (version 0.2.1), an RNA velocity-based lineage inference tool [4]. Markov simulations were initialized at random cells with a root cell probability above 0.99. Leiden clusters 0 and 3 (Fig 2B), corresponding to the terminally exhausted and memory-like terminal states, respectively, were chosen as endpoints for the Markov chain sampling according to biological knowledge and their relatively higher endpoint probability. The required number of simulations per endpoint was automatically adjusted based on the dataset size, leading to 13,422 initial simulations overall, to generate a minimum of 826 samples per endpoint. The number of required steps was set to 79 in an automated procedure. Briefly, the initial state probability distribution, obtained from root cell probabilities, was iteratively multiplied with the transition probability matrix for 1,000 iterations. The number of steps was chosen by convergence, i.e., the step where the change of the difference to the stationary distribution obtained from endpoint probabilities was below a predefined threshold. All simulated paths were aligned per endpoint using dynamic time warping and clustered with HDBSCAN using Hausdorff distances to obtain aggregated trajectories. Trajectory coordinates were determined as the mean coordinates of the aligned sequences. Neighboring cells were aligned to the trajectories using an alignment score, computed as the cosine similarity between a trajectory segment and the average of the cell's transition vector to its neighboring cells. The final alignment score of a cell towards a trajectory is the average over all trajectory segments with a non-zero alignment

score [4]. The average step and cell fate probability for each cell and trajectory were computed based on the alignment scores.

**Exploratory trajectory inference.** Simulations of cell sequences were sampled and aggregated into trajectories using a reimplementation of Cytopath for exploratory trajectory inference within the cy2path package (version 0.0.3) [37]. We sampled 1,000 Markov chains of cells, starting from random cells chosen according to their root cell probabilities. The required number of steps was estimated, as described above, by iterative multiplication of the transition probability matrix with the initial state probability distribution. Here, the stationary state distribution was directly estimated from the eigenvectors of the transition probability matrix. Furthermore, the default threshold on the difference to the stationary distribution is lower in cy2path than in Cytopath, resulting in simulations with more steps. This approach enables greater sensitivity in detecting transitions near high end point probability regions. The number of steps for Markov chain sampling was set to 228 based on convergence.

To cluster sampled cell sequences, a distance matrix between all sequences was computed using the dynamic time warping algorithm, which aligns temporal sequences that vary in speed. The alignment of two simulated cell sequences is based on the Euclidean distance of individual cells in PCA space and yields a distance measure between the complete sequences. With this distance measure, agglomerative, hierarchical clustering with Ward linkage was used to obtain a linkage tree. Clusters corresponding to lineages were obtained by cutting the tree at the desired depth. When the number of lineages is not known *a priori*, silhouette coefficients for different cutoffs are computed based on the same dynamic time warping distances between simulated cell sequences. The clustering with the highest silhouette coefficient is chosen, and its clusters are reported as the inferred lineages. Trajectory coordinates were determined as the median coordinate of all sequences in the respective cluster.

In summary, the main methodological differences of exploratory trajectory inference compared to the Cytopath approach described in [4], enabling a more unbiased analysis, are as follows. Briefly, (1) end states are not explicitly specified. Instead, a fixed number of simulations is performed, with the number of steps determined through convergence to the stationary state distribution. In contrast, Cytopath aims to sample chains that reach the specified end points. (2) Trajectories are inferred based on these simulations by hierarchical, agglomerative clustering with dynamic time warping distance instead of clustering with HDBSCAN using Hausdorff distances. In principle, agglomerative clustering enables the analysis of the cluster hierarchy to obtain trajectories at different granularities, in particular for complex differentiation processes. Dynamic time warping distance was chosen instead of Hausdorff distances because it offers a more intuitive approach to compare cell sequences that are not temporally synchronized. Furthermore, Cytopath also uses dynamic time warping to align sequences before computing Hausdorff distances between them. Thus, we chose the more straightforward approach of directly using dynamic time warping distance for exploratory trajectory inference.

**Uncertainty estimation of inferred trajectories.** To estimate the uncertainty of inferred trajectories, we performed exploratory trajectory inference 10 times on the same data with different random seeds. Coordinates of inferred lineages in the UMAP embedding were computed as described above. Since the order of lineages depends on the random simulations, particularly for higher numbers of lineages, we mapped the inferred lineages to their closest matches across runs. To this end, the lineages from the first run were chosen as the reference. Then, for each of the other runs, each lineage was mapped to the closest reference lineage in terms of dynamic time warping distance in PCA space. The mean and standard deviation of trajectory coordinates in UMAP space, as well as the variance of coordinates in PCA

space across the repeated mapped runs, were used to visualize the uncertainty of inferred lineages on the UMAP embedding. For the latter, we visualized the sum of the variance of all 50 principal components.

**Monocle3.** Monocle3 [9,23] (version 1.3.5) was run by executing the functions `cluster_cells`, `learn_graph`, and `order_cells`. For clustering the cells, the parameter `reduction_method` was set to "UMAP". All cells in Leiden cluster 5 were set as root cells for ordering the cells. All other parameters were set to default values.

**Slingshot.** Slingshot [8] (version 2.4.0) was run on the UMAP embedding and Leiden clusters with default parameters. Cluster 5 was used as the start cluster, while clusters 0 and 3 were specified as end clusters.

**CellRank.** A CellRank [6] (version 2.0.7) analysis was performed using the `VelocityKernel`, initialized from the processed data. A transition probability matrix was computed with the "stochastic" model. The GPCCA estimator [38] was fitted on the data, using Leiden clusters and searching for the optimal number of macrostates in the range between 4 and 20. Macrostates corresponding to terminally exhausted and memory-like cells were set as the terminal states and used to compute cell fate probabilities.

**VIA.** To run trajectory inference with VIA [24] (pyVIA version 0.2.4), the model was initialized from the PCA projection of the data, using Leiden clusters as the labels. Log-transformed counts and the RNA velocity matrix were also provided. The model was run with default parameters. Trajectory curves were computed with the `plot_trajectory_curves` function.

**Run time and memory scaling study.** Processed scRNAseq data, including spliced and unspliced counts, of the mouse gastrulation process [25] were obtained from scVelo [14] (version 0.3.3). Genes with less than 20 shared counts (spliced and unspliced) were removed. The expression matrix, spliced, and unspliced counts were normalized per cell to the respective median total counts per cell. Highly variable genes were computed in scVelo using the flavor "seurat". The gene expression matrix was log-transformed. A neighborhood graph with 30 nearest neighbors was computed based on the PCA projection obtained from the preprocessed data. Gene moments of spliced and unspliced counts were estimated using 30 nearest neighbors. RNA velocities were calculated with the "stochastic" model, and a velocity graph was computed using nearest-neighbor connectivities. Terminal states and a transition probability matrix between cells were computed using scVelo, disabling self-transitions.

Random subsets of the data were sampled, containing 5,000, 10,000, 20,000, and 50,000 cells. For each subset, we repeated the steps described above from the computation of nearest neighbors to the transition probability matrix. Exploratory trajectory inference was run five times on all subsets and the complete data with different numbers of simulated Markov chains (100, 500, 1,000, and 2,000). We evaluated run times and memory requirements on a compute cluster with AMD EPYC 7343 16-Core processors, allocating 32 cores for each run and setting the maximum allowed memory usage to 64 GB.

**Simulation of positive control data.** To generate positive control data with simple, well-known ground truth trajectories of varying topology, we constructed trajectories in two dimensions (see left column in S19 Fig). To this end, we defined cluster means forming the desired topology and sampled 100 points from a bivariate Gaussian distribution around the means. Additionally, we sampled interpolated points lying between the main clusters (100 points for each pair of consecutive clusters). This resulted in 1,300, 3,700, 2,000, and 2,400 data points for the bifurcating, double bifurcating, converging, and cyclical topologies, respectively. Velocities were defined in two dimensions to point into the general direction of the next cluster along the trajectory, with small additional Gaussian noise. A projection matrix

was sampled from a standard normal distribution and used to project points and velocities from two into 100 dimensions. Gaussian noise was added to the high-dimensional data points. Then, the data was processed with the typical steps, using Scanpy and scVelo: A PCA was computed, a neighborhood graph with 30 nearest neighbors was determined, and a UMAP embedding was generated. A velocity graph was built using neighbor connectivities. Terminal states and a transition matrix were computed.

Additional positive control datasets were generated with dyngen [29] in R using the bifurcating, cycle, consecutive bifurcating, trifurcating, converging, and disconnected backbones. For each dataset, 500 genes were simulated for 5,000 cells. The number of transcription factors was set to the number of modules in the respective backbone. Half of the remaining genes were simulated as targets, the rest as housekeeping genes. The simulation parameters were set as follows: `total_time = 1000`, `census_interval = 2`, `ssa_algorithm = ssa_etl(tau = 300 / 3600)`. We performed wild-type simulations with 500 simulations for a snapshot experiment, computing RNA velocity as well. The data was further processed using Scanpy and scVelo, using the simulated counts, spliced counts, unspliced counts, and RNA velocity. Genes with less than 20 shared counts (spliced and unspliced) were removed, and the three count matrices were normalized to the respective median of total counts. The expression data was log-transformed and scaled. PCA, a neighborhood graph, and a UMAP projection were computed with default parameters. Normalized spliced and unspliced counts were smoothed using `scvelo.pp.moments`, and a velocity graph was built using neighbor connectivities. Terminal states and a transition matrix were computed.

For each simulated dataset, we performed exploratory trajectory inference using cy2path with automatic detection of the number of lineages based on silhouette coefficients.

**Preparation of negative control data.** A dataset of peripheral blood mononuclear cells (PBMCs) [30] was obtained from scVelo [14] and processed using Scanpy [35] and scVelo. Genes with less than 20 shared counts (spliced and unspliced) were removed and 2,000 highly variable genes were computed using the scVelo function `filter_and_normalize`. The data was scaled, and PCA was performed. A neighborhood graph with 30 nearest neighbors was computed. Moments of spliced and unspliced counts were computed. A UMAP embedding was generated with default parameters. RNA velocities were estimated using the "stochastic" model, a velocity graph was estimated using neighborhood connectivities, and terminal states and a transition probability matrix were computed. Exploratory trajectory inference was performed with cy2path.

### Adoptive transfer experiments

After 5 days of chronic infection, CD8 T cell-enriched samples from the spleen were stained with α-CD8-PerCP and α-CXCR6-PE and α-CD45.1-APC or α-CD45.1-FITC if in combination with GFP expression (reporting TCF1 activity) to sort P14 cells into exhausted, memory-like and pre-committed populations (ARIA cell sorter, BD Biosciences).

Sorted exhausted ($10^6$ cells), memory-like ($2 \cdot 10^5$ cells), and pre-committed ($5 \cdot 10^4$ cells) cells were transferred via intravenous (IV) injection into infection-matched hosts infected with Clone-13. Cells were recovered from the spleens of these mice 12 days post-infection prior to phenotypic characterization.

### Flow cytometry

Surface staining was performed at room temperature for 30 min in FACS buffer (2% FCS, 1% EDTA in PBS). LIVE/DEAD™ Fixable Near-IR (Thermo Fisher) was used to discriminate alive from dead cells. Fluorophore-conjugated antibodies used for flow cytometry were purchased

from BioLegend (Lucerna Chem AG) (α-CD45.1 BV711 A20; α-CD45.1 APC A20; α-CD8 PerCP 53-6.7; α-CD8 BV395 53-6.7; α-PD-1 PE-BV605 29F.1A12; α-CXCR6 PE SA051D1; α-CD8 BV395 53-6.7). Data was acquired with an LSR II Fortessa using Diva software (BD Biosciences). Analysis, gating, and plotting were done using FlowJo (BD Biosciences).

## Supporting information

**S1 Fig. Expression of relevant genes on a UMAP embedding.**
(TIF)

**S2 Fig. Composition of phenotypic groups with respect to samples.** (A) Cell numbers per group for each sample time point. (B) Cell numbers per sample time point for each phenotypic group. (C) Fractional composition per sample time-point. (D) Fractional composition per phenotypic group.
(TIF)

**S3 Fig. Cell cycle scoring.** The early and proliferating subpopulations were classified as a mix of G2/M- and S-phase cells.
(TIF)

**S4 Fig. Top 15 differentially expressed genes between relevant populations.**
(TIF)

**S5 Fig. Top 20 differentially expressed genes between the two differentiation endpoints.**
(TIF)

**S6 Fig. Boxenplots of gene expression along Cytopath pseudotime.** Genes with differential expression patterns between exhausted and memory-like trajectories are shown.
(TIF)

**S7 Fig. Lineages obtained through hierarchical clustering of Cytopath simulations.** (A–C) Three, four, and five lineages are obtained by cutting the hierarchical tree at the corresponding height. Insets show the number of simulations per lineage. Each lineage is shown in a distinct color. (D-E) Average step for cells in the exhausted (D) and memory-like trajectory (E). (F–G) Cell fate probabilities along the exhausted (F) and memory-like trajectory (G).
(TIF)

**S8 Fig. Trajectory inference with Monocle3, Slingshot, CellRank, and VIA.** (A) Trajectories and pseudotime inferred by Monocle3 [9,23]. (B) Trajectories and pseudotime inferred by Slingshot [8]. (C–D) CellRank [6] fate probabilities for the terminally exhausted (C) and memory-like endpoint (D). (E) Trajectories and pseudotime inferred by VIA [24].
(TIF)

**S9 Fig. Stability of inferred trajectories across multiple runs.** (A–F) Mean and standard deviation of inferred trajectory coordinates across 10 runs for different numbers of inferred lineages. Each lineage is represented by a distinct color.
(TIF)

**S10 Fig. Uncertainty of inferred trajectories across multiple runs.** (A–F) Mean of inferred trajectory coordinates across 10 runs for different numbers of inferred lineages. The color scale represents the uncertainty in the form of the summed variance of 50 principal components across the runs.
(TIF)

**S11 Fig. Silhouette coefficients for different numbers of lineages.** Colors represent individual simulation runs.
(TIF)

**S12 Fig. Run time and memory requirements of exploratory trajectory inference.** (A-B) Run time (A) and memory (B) requirements plotted versus the number of cells in the dataset, split by the number of simulated chains. Bars show the mean of five replicates. Error bars correspond to the standard deviation. (C) For a fixed number of chains (1,000), the run time rises approximately linearly with the number of cells. (D) For a fixed dataset size (20,000 cells), run time increases quadratically with the number of simulated chains. (E) Run time in relation to the number of simulated steps for a fixed number of chains (1,000). Lines and shaded regions in panels C-E show the mean and standard deviation of five replicates. (F) Relation between the number of cells in the dataset and the number of required simulation steps.
(TIF)

**S13 Fig. Root cell probabilities for T cell exhaustion data based on RNA velocities computed with varying parameters.** (A) Velocities computed with the deterministic model. (B) Velocities computed with the stochastic model. (C) Velocities computed with the dynamical model. Columns correspond (from left to right) to 15, 30, 50, and 100 nearest neighbors used for velocity computation. All subpanels show the inferred, embedded velocities smoothed on a grid.
(TIF)

**S14 Fig. End point probabilities for T cell exhaustion data based on RNA velocities computed with varying parameters.** (A) Velocities computed with the deterministic model. (B) Velocities computed with the stochastic model. (C) Velocities computed with the dynamical model. Columns correspond (from left to right) to 15, 30, 50, and 100 nearest neighbors used for velocity computation. All subpanels show the inferred, embedded velocities smoothed on a grid.
(TIF)

**S15 Fig. Correlation of terminal state probabilities based on RNA velocities computed with varying parameters for T cell exhaustion data.** (A) Pairwise Spearman's correlation between root cell probabilities. (B) Pairwise Spearman's correlation between end point probabilities. Comparisons were made between the three modes of RNA velocity inference (deterministic, stochastic, and dynamical), and four different parameters for the number of neighbors (15, 30, 50, and 100 nearest neighbors).
(TIF)

**S16 Fig. Lineages inferred with exploratory trajectory inference for T cell exhaustion data based on RNA velocities computed with varying parameters.** (A) Lineages based on velocities computed with the deterministic model. (B) Lineages based on velocities computed with the stochastic model. (C) Lineages based on velocities computed with the dynamical model. Columns correspond (from left to right) to 15, 30, 50, and 100 nearest neighbors used for velocity computation. Each panel shows the mean of inferred trajectory coordinates across 10 runs. The color scale represents the uncertainty in the form of the summed variance of 50 principal components across the runs.
(TIF)

**S17 Fig. Inferred trajectories for T cell exhaustion data with different simulation lengths.** (A-F) Mean and standard deviation of inferred trajectory coordinates across 10 runs for different numbers of simulation steps. Each lineage is represented by a distinct color.
(TIF)

**S18 Fig. Trajectory inference for T cell exhaustion data using a connectivity-based transition matrix.** (A) Root cell probabilities based on connectivities. (B) End point probabilities based on connectivities. (C) Inferred trajectories based on the connectivity transition matrix.
(TIF)

**S19 Fig. Trajectory inference for simulated data with known ground-truth trajectories.** Data was generated in two dimensions with a bifurcating (A), double bifurcating (B), converging (C), and cyclical (D) topology. The left column shows the original, two-dimensional data and velocities, with points colored by their cluster of origin. The center column depicts velocities projected onto a UMAP embedding, with points colored by cluster. The right column shows lineages inferred with exploratory trajectory inference. Points are colored by their root cell probabilities to indicate trajectory directionality.
(TIF)

**S20 Fig. Trajectory inference for dyngen-simulated data with known ground-truth trajectories.** (A-C) Trajectory inference for dyngen's bifurcating backbone. Velocities were projected onto a UMAP embedding, colored by root cell (A) and end point (B) probability. Inferred lineages are shown on the same embedding, colored by true simulation time (C). (D-F) Trajectory inference for dyngen's cycle backbone. Velocities were projected onto a UMAP embedding, colored by root cell (D) and end point (E) probability. Inferred lineages are shown on the same embedding, colored by true simulation time (F). (G-J) Inferred lineages for the consecutive bifurcating (G), trifurcating (H), converging (I), and disconnected (J) dyngen backbone. Color represents ground-truth simulation time.
(TIF)

**S21 Fig. Trajectory inference for PBMCs as a negative control.** (A) Root cell probabilities based on RNA velocity. (B) End point probabilities based on RNA velocity. (C) Cell type annotations. (D) Lineages inferred with exploratory trajectory inference.
(TIF)

**S22 Fig. Distribution of root cell and end point probabilities for various datasets.** Joint and marginal distributions of root cell and end point probabilities for T cell exhaustion (A), PBMC (B), and dyngen cycle data (C). The black rectangle indicates cells with both nonnegligible root cell and end point probability, with the percentage of cells falling into this area indicated in the top right corner. For the marginal distributions, median absolute deviation (MAD) is shown.
(TIF)

## Acknowledgments

We thank Franziska Wagen and Nathalie Oetiker for great technical support. We are grateful for the constructive input of the members of the Claassen, Oxenius, Joller, and Sallusto Group during discussions and group meetings. The authors thank the International Max Planck Research School for Intelligent Systems (IMPRS-IS) for supporting J.T.S.

## Author contributions

**Conceptualization:** Revant Gupta, Dario Cerletti, Ioana Sandu, Annette Oxenius, Manfred Claassen.

**Formal analysis:** Jan T. Schleicher, Revant Gupta, Manfred Claassen.

**Funding acquisition:** Manfred Claassen.

**Investigation:** Jan T. Schleicher, Revant Gupta, Dario Cerletti, Ioana Sandu, Annette Oxenius, Manfred Claassen.

**Methodology:** Jan T. Schleicher, Revant Gupta, Dario Cerletti, Manfred Claassen.

**Software:** Jan T. Schleicher, Revant Gupta.

**Supervision:** Annette Oxenius.

**Validation:** Dario Cerletti, Ioana Sandu.

**Writing – original draft:** Revant Gupta, Dario Cerletti, Annette Oxenius, Manfred Claassen.

**Writing – review & editing:** Jan T. Schleicher, Revant Gupta, Dario Cerletti, Ioana Sandu, Annette Oxenius, Manfred Claassen.

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
