## [Decision Letter · Decision Letter 0]

21 Apr 2025

PONE-D-25-14715Exploratory trajectory inference reveals convergent lineages for CD8 T cells in chronic LCMV infectionPLOS ONE

Dear Dr. Claassen,

Thank you for submitting your manuscript to PLOS ONE. After careful consideration, we feel that it has merit but does not fully meet PLOS ONE’s publication criteria as it currently stands. Therefore, we invite you to submit a revised version of the manuscript that addresses the points raised during the review process.

We look forward to receiving your revised manuscript.

Kind regards,

Xianmin Zhu

Academic Editor

PLOS ONE

“This work was supported by the ETH Z üurich (grant no. 470

ETH-39 14-2 to M.C. and A.O.) and the Novartis Foundation for Biomedical Research, 471

DFG CL 792/1-1 and the Center for Personalized Medicine (ZPM) and DFG EXC 2180.”

“We thank Franziska Wagen and Nathalie Oetiker for great technical support. We are 466

grateful for the constructive input of the members of the Claassen, Oxenius, Joller 467

and Sallusto Group during discussions and group meetings. The authors thank the 468

International Max Planck Research School for Intelligent Systems (IMPRS-IS) for 469

supporting J.T.S. Funding: This work was supported by the ETH Z¨urich (grant no. 470

ETH-39 14-2 to M.C. and A.O.) and the Novartis Foundation for Biomedical Research, 471

DFG CL 792/1-1 and the Center for Personalized Medicine (ZPM) and DFG EXC 2180.”

“This work was supported by the ETH Z üurich (grant no. 470

ETH-39 14-2 to M.C. and A.O.) and the Novartis Foundation for Biomedical Research, 471

DFG CL 792/1-1 and the Center for Personalized Medicine (ZPM) and DFG EXC 2180.”

4. We notice that your supplementary figures are included in the manuscript file. Please remove them and upload them with the file type 'Supporting Information'. Please ensure that each Supporting Information file has a legend listed in the manuscript after the references list.

Reviewers' comments:

Reviewer's Responses to Questions

**Comments to the Author**

1. Is the manuscript technically sound, and do the data support the conclusions?

Reviewer #1: Yes

Reviewer #2: Yes

Reviewer #3: No

2. Has the statistical analysis been performed appropriately and rigorously? 

Reviewer #1: Yes

Reviewer #2: Yes

Reviewer #3: No

3. Have the authors made all data underlying the findings in their manuscript fully available?

Reviewer #1: Yes

Reviewer #2: Yes

Reviewer #3: Yes

4. Is the manuscript presented in an intelligible fashion and written in standard English?

Reviewer #1: Yes

Reviewer #2: Yes

Reviewer #3: Yes

5. Review Comments to the Author

Reviewer #1: In this paper, Revant proposes a new method called Cytopath to infer trajectories in scRNA-seq data by leveraging RNA velocity information. They present an in-depth analysis of a chronic LCMV infection dataset and support their findings with flow cytometry experiments. The proposed method appears promising, and the accompanying biological analyses are compelling. However, I have a few concerns:

1. Some aspects of Cytopath remain unclear. For instance, the hierarchical clustering based on the simulated sequences is not well explained. What features are used in the clustering—only gene expression, or also information derived from the simulated trajectories? Does Cytopath require prior knowledge such as cell type or state labels? How does the method determine or control the number of inferred trajectories? Clarifying these points in the Results and Methods sections would greatly aid readers in understanding the approach.

2. It is well known that different RNA velocity inference methods can yield substantially different results [1]. It would be helpful if the authors could explore how such variation in RNA velocity affects the performance and output of Cytopath.

3. The stability of Cytopath needs further discussion. Since the method involves simulation, it would be important to know whether multiple runs on the same dataset yield consistent trajectories.

4. Finally, a discussion on Cytopath’s computational efficiency—including time and memory usage—would be valuable for potential users.

References:

[1] Zheng, S.C., Stein-O’Brien, G., Boukas, L. et al. Pumping the brakes on RNA velocity by understanding and interpreting RNA velocity estimates. Genome Biol 24, 246 (2023). https://doi.org/10.1186/s13059-023-03065-x

Reviewer #2: The authors proposed a novel framework termed exploratory trajectory inference to address limitations in resolving complex lineage topologies, particularly convergent differentiation trajectories, from single-cell RNA sequencing (scRNA-seq) data. By applying this approach to an original dataset of CD8 T cell responses in chronic LCMV infection, the authors identified several distinct yet converging developmental paths to T cell exhaustion. The inferred trajectories are experimentally supported through adoptive transfer studies. I find the framework proposed in the paper interesting, and the experimental validation is convincing. However, I think the authors should provide more clarification regarding the advances over their previous studies, as well as include additional simulation benchmarking results to better demonstrate the performance of the approach.

Here are my specific comments:

1. The proposed framework, exploratory trajectory inference, appears to be based on cy2path (Ref. 35), which seems closely related to Cytopath (Ref. 14). Given the strong methodological overlap among these works, could the authors clarify how the current manuscript advances beyond these prior studies? Specifically, it would be helpful to delineate which aspects of the methodology are novel in this manuscript, and how it extends or differs from cy2path and/or Cytopath. For example, the description in the Exploratory Trajectory Inference subsection of the Methods section could more clearly highlight the conceptual differences.

2. RNA velocity methods like scVelo are based on abundance quantification and involve numerous steps and parameters, which can lead to variability in results (https://doi.org/10.1371/journal.pcbi.1008585; https://doi.org/10.1371/journal.pcbi.1010492). This variability can propagate downstream through the state transition matrix, potentially affecting trajectory inference outcomes. Can the authors provide some explanation or further analysis on whether exploratory trajectory inference is sensitive to these factors?

3. The authors mention around line 180 that "we did not enforce simulations to reach predefined endpoints. Instead, we simulated 1000 cell state sequences with a predetermined, automatically chosen number of steps," and on line 413 that "The number of steps for Markov chain sampling was set to 228 based on convergence." I am a little confused about how the number of steps was determined. Could the authors clarify how both the number of sequences and the number of steps were chosen, and provide analysis on whether exploratory trajectory inference is sensitive to these parameters?

4. Can this framework handle cyclical processes? Please clarify which types of topologies the proposed method is capable of inferring.

5. Does the computational cost of exploratory trajectory inference increase more than linearly with the number of cells? If so, could the authors evaluate the computational efficiency of the method on datasets with larger cell numbers?

6. Minor points about the figures:

a. Please check whether the title of Figure 1 is correct.

b. On line 99, should the reference to Fig. 1C be Fig. 2B instead?

c. On lines 210, 214, 218, and 223, the word "Fig" appears to be missing.

Reviewer #3: In their study, “Exploratory trajectory inference reveals convergent lineages for CD8 T cells in chronic LCMV infection”, the authors describe a methodology for inferring trajectories between cells in scRNA-seq data by aggregating simulations of cell state transition dynamics without an imposed external structure/geometry, to propose potentially novel trajectories. They apply this methodology, termed ‘exploratory trajectory inference’, to time-series scRNA-seq data and highlight convergent cell state transitions where terminally exhausted CD8 T cells can develop from memory-like or non-memory-like exhausted CD8 T cells.

Overall, the concept of using ensembles of simulated cell state transitions for more generalizable trajectory inference is interesting but the study is lacking simulated or data-based controls as well as other measures of accuracy or robustness to assess the properties of the proposed methodology. Without such baselines and applications, it is difficult to understand how one should apply this approach and interpret or trust the results. The experimental validation, though interesting, does not address the capabilities of the analysis methods themselves, e.g., if or how easily ‘incorrect’ or nonexistent paths could be produced in the analysis even if ‘correct’ paths also appear.

Major Comments:

1. A major issue in RNA velocity analyses is the impact of the chosen dimension reduction/embedding method on the inferred velocity values and transition probabilities [1,2]. This can lead to bias and/or arbitrary final velocity values warped by the placement of cells in the 2D visualizations. Thus it is necessary to understand if the UMAP embedding used here affects the resulting trajectories in any way.

- Both the hyperparameters of the UMAP algorithm and the choice of k-nearest neighbors to smooth over in other steps of the velocity pipeline (such as calculating the gene moments) impact the definition of cell neighborhoods and their placement in 2D [1,2]. Thus the trajectory inference results should be shown to be robust to a range of reasonable hyperparameters. Likewise, cells at particular times or types can be removed to see if arbitrary trajectories are induced based on this change in locations of cells in embedding space.

2. As the authors describe in the Cytopath paper [3], the use of RNA velocity values is not necessary to construct the transition matrix used, which highlights a control/baseline that is missing from here. The same cosine distance/similarity calculated between velocity vectors can be calculated between cells (preferably using both unspliced and spliced counts as this is the information used in velocity) to create a transition matrix not reliant on inferred velocity values which provides a point of comparison for both the broader utility of this approach and the potential instabilities induced by the RNA velocity inference pipeline.

3. The output of the exploratory inference methodology additionally provides no metrics of uncertainty or confidence in the inferred trajectories, making it difficult to assess the believability of potentially novel paths. For example, obtaining statistics on the variance of the various trajectory coordinates would be helpful as these are then averaged over for the final trajectories visualized. The definitions of trajectory 'segments' and 'coordinates' can also be further clarified (e.g., what calculations are used to obtain 'segments', what space/dimension are these in)?

- A scrambled transition matrix may also provide a baseline for comparison/confidence.

- Potentially the frequency of a particular path is also something that could be visualized and used as a qualitative assessment.

4.Generally the question of how spurious the trajectories inferred by this exploratory methodology are is not addressed, beyond the comparison of trajectories obtained by cutting the hierarchical clustering dendrogram at different levels. To demonstrate if the method is relatively robust and accurate, comparisons need to be made to control settings (potentially through simulations as well as data) which encompass settings without continuous relationships between cells (negative control) and settings with 'ground truth' trajectories. These will demonstrate how easily trajectories are inferred when they should or shouldn't be present, providing a baseline and external context for the result on the CD8 T cells.

5.As currently written, it is unclear what the advantage or novel finding of the cy2path/‘exploratory’ methodology (Fig. 5C,D) is over the Cytopath results in Fig 4A. In Fig. 4A the path from early to exhausted memory T cells to terminally exhausted cells is present in the results obtained from Cytopath analysis, with the width of the arrows potentially suggesting more or less likely transitions. The proposed exploratory method, following the Cytopath section, shows the same trajectories (Fig. 5C,D), but without any width differences, i.e., no visual difference in how 'likely' either trajectory is. It seems that both approaches show this main finding, and thus it is unclear what the cy2path/exploratory approach adds in terms of novel path discovery.

- The study should also compare to other existing methods that use transition probability matrices or random walks to infer trajectories between cells (e.g., CellRank, VIA) [4,5].

Minor Comments:

1. There are several statements that refer to validation of 'continuous' relationships between cells based on the qualitative UMAP space, which is known to create or distort quantitative cell-cell relationships [1,2,6]. Thus this space does not qualify as a measure of continuity or a validation of a real developmental lineage.

2. Two different clustering approaches are used on the outputs of the Cytopath and the cy2path/exploratory methodologies. It would be good to see a rationale for why different approaches were used in these cases.

References:

1. Zheng, S. C., Stein-O’Brien, G., Boukas, L., Goff, L. A. & Hansen, K. D. Pumping the brakes on RNA velocity by understanding and interpreting RNA velocity estimates. Genome Biol. 24, 246 (2023).

2. Gorin, G., Fang, M., Chari, T. & Pachter, L. RNA velocity unraveled. PLoS Comput. Biol. 18, e1010492 (2022).

3. Gupta, R., Cerletti, D., Gut, G., Oxenius, A. & Claassen, M. Simulation-based inference of differentiation trajectories from RNA velocity fields. Cell Rep. Methods 2, 100359 (2022).

4. Lange, M. et al. CellRank for directed single-cell fate mapping. Nat. Methods 19, 159–170 (2022).

5. Stassen, S. V., Yip, G. G. K., Wong, K. K. Y., Ho, J. W. K. & Tsia, K. K. Generalized and scalable trajectory inference in single-cell omics data with VIA. Nat. Commun. 12, 5528 (2021).

6. Kharchenko, P. V. The triumphs and limitations of computational methods for scRNA-seq. Nat. Methods 18, 723–732 (2021).

6. PLOS authors have the option to publish the peer review history of their article (what does this mean?). If published, this will include your full peer review and any attached files.

Reviewer #1: No

Reviewer #2: No

Reviewer #3: No

---

## [Author Response · Author response to Decision Letter 1]

10 Jun 2025

Please find below the point-by-point response to the reviewer comments. Reviewer comments are copied in green font, author responses in black font, and text citations from the revised manuscript are displayed in black italic font. Reference numbers within the cited text refer to the manuscript references.

Reviewer #1: In this paper, Revant proposes a new method called Cytopath to infer trajectories in scRNA-seq data by leveraging RNA velocity information. They present an in-depth analysis of a chronic LCMV infection dataset and support their findings with flow cytometry experiments. The proposed method appears promising, and the accompanying biological analyses are compelling. However, I have a few concerns:

1. Some aspects of Cytopath remain unclear. For instance, the hierarchical clustering based on the simulated sequences is not well explained. What features are used in the clustering—only gene expression, or also information derived from the simulated trajectories?

As the reviewer rightly pointed out, the clustering was not explained sufficiently in the manuscript. To clarify this, we added a more detailed description of the hierarchical clustering step in the methods section under “Exploratory Trajectory Inference”. We cite the corresponding text here (lines 523-529):

To cluster sampled cell sequences, a distance matrix between all sequences was computed using the dynamic time warping algorithm, which aligns temporal sequences that vary in speed. The alignment of two simulated cell sequences is based on the Euclidean distance of individual cells in PCA space, yielding a distance measure between the complete sequences. With this distance measure, agglomerative, hierarchical clustering with Ward linkage was used to obtain a linkage tree. Clusters corresponding to lineages were obtained by cutting the tree at the desired depth.

Does Cytopath require prior knowledge such as cell type or state labels?

Thank you for the question. Cytopath and cy2path do not require prior knowledge of cell types or state labels. However, clusters can be used to guide the analysis of the original cytopath method by predefining root or end clusters. In contrast, the novel approach of exploratory trajectory inference, using the cy2path reimplementation of cytopath with hierarchical clustering, always uses root cell and end point probabilities on a per-cell level, which can also be used in the original cytopath. Both approaches use cell clusters in the downstream analysis to align cells to trajectories and infer their differentiation fate. The prerequisites of exploratory trajectory inference are now described in the discussion section (lines 293-299):

Exploratory trajectory inference does not require prior knowledge of cell types or state labels. The only prerequisite for performing simulations is the presence of a transition probability matrix and root and end state probabilities. In the absence of these, they are computed on the spot based on RNA velocity estimates, if available.

For downstream analysis, including alignment of cells to trajectories to infer pseudotime and cell fate, cluster information is required. Cell type annotations can aid the biological interpretation of inferred trajectories.

How does the method determine or control the number of inferred trajectories?

We thank the reviewer for this comment, which prompted us to implement an unbiased approach for determining the best number of inferred trajectories. The methods section was updated accordingly to describe how silhouette coefficients can be used to this end (lines 529-533):

When the number of lineages is not known a priori, silhouette coefficients for different cutoffs are computed based on the same dynamic time warping distances between simulated cell sequences. The clustering with the highest silhouette coefficient is chosen, and its clusters are reported as the inferred lineages.

The silhouette coefficients for different numbers of lineages are shown in S10 Fig.

2. It is well known that different RNA velocity inference methods can yield substantially different results [1]. It would be helpful if the authors could explore how such variation in RNA velocity affects the performance and output of Cytopath.

We performed an additional analysis to investigate how differences in RNA velocity estimates influence our exploratory trajectory inference approach. To this end, we qualitatively compared inferred trajectories across a range of values for the number of neighbors used for the estimation of RNA velocity, as well as for all three velocity modes implemented in scVelo.

We show the results in S12 Fig, S13 Fig, and S14 Fig and describe them in the Discussion section. In brief, the inferred trajectories are qualitatively stable across the range of tested parameters. In particular, for the stochastic model of RNA velocity, the inferred trajectories are highly consistent across the different values of the number of neighbors, with the only difference being the failure to identify the transition from memory-like to terminally exhausted T cells when the number of neighbors is at the lower end.

We added the following specific text in the Discussion section (lines 343-360):

Computational approaches such as the proposed approach require users to set parameters. We find that exploratory trajectory inference is robust over a wide range of parameter choices. For instance, RNA velocity inference is sensitive to changes in the parameters of the chosen inference method. We assessed the influence of different velocity estimation procedures on exploratory trajectory inference.

For the three different models in scVelo (“deterministic”, “stochastic”, and “dynamical”) and four different numbers of neighbors (15, 30, 50, and 100), the inferred velocities, root cells (S12 Fig), and end points (S13 Fig) were very similar and mostly differed in the position of the root cells within the early cluster and in the magnitude of end cell probabilities of the terminally exhausted and memory-like clusters. Further, the lineages inferred with exploratory trajectory inference (S14 Fig) were also stable across the different parameter settings. Notably, the only qualitative difference between the trajectories inferred from stochastic velocities was the failure to detect the transition from memory-like to terminally exhausted cells with 15 neighbors. The dynamical model resulted in larger deviations for the extreme values of the number of neighbors (15 and 100), but produced similar qualitative trajectories for 30 and 50 neighbors. Our findings reported in this study, obtained with the stochastic model and 30 nearest neighbors, were supported by the vast majority of trajectory inference analyses with other parameters.

3. The stability of Cytopath needs further discussion. Since the method involves simulation, it would be important to know whether multiple runs on the same dataset yield consistent trajectories.

To address this valid concern about the stability of exploratory trajectory inference, we performed 10 separate runs on the considered dataset and compared the results. The inferred lineages were consistent across the ten runs, with only one single run missing the transition from the memory-like to the terminally-exhausted state (see figure below; SC stands for the silhouette coefficient of the respective clustering).

To further quantify this stability, we computed the mean and standard deviation of trajectory lineage coordinates across the repeated runs. This procedure is described in the methods section (lines 551-560):

To estimate the uncertainty of inferred trajectories, we performed exploratory trajectory inference 10 times on the same data with different random seeds.

Coordinates of inferred lineages in the UMAP embedding were computed as described above. Since the order of lineages depends on the random simulations, particularly for higher numbers of lineages, we mapped the inferred lineages to their closest matches across runs. To this end, the lineages from the first run were chosen as the reference. Then, for each of the other runs, each lineage was mapped to the closest reference lineage in terms of dynamic time warping distance in PCA space. The mean and standard deviation of trajectory coordinates in UMAP space across the repeated mapped runs were used to visualize the uncertainty of inferred lineages on the UMAP embedding.

The results of the analysis are included in S9 Fig of the manuscript and described in the results section (lines 215-222):

To assess the stability of inferred trajectories, we repeated the Markov chain simulations 10 times. In particular, in the case of two lineages, the results were stable, with a low deviation of inferred trajectory coordinates around the mean across all runs (S9 Fig). We found increasing uncertainty and overlap between lineages with an increasing number of lineages. Additionally, we computed silhouette coefficients for the different numbers of lineages (S10 Fig). Together with the increasing uncertainty, these results indicate that the best number of lineages (with the highest silhouette coefficient) for this dataset is two.

Rebuttal Figure 1: Inferred trajectories in repeated runs on the same dataset (arbitrary trajectory coloring across panels).

4. Finally, a discussion on Cytopath’s computational efficiency—including time and memory usage—would be valuable for potential users.

We recognize the importance of run time and memory evaluations for potential users of our approach, highlighted by the reviewer. Accordingly, we have added an analysis of computational efficiency based on subsets of a dataset with different numbers of cells. Furthermore, we tested different settings for the number of simulations. The results of this analysis are shown in S11 Fig and described in the results section (lines 238-248):

Finally, to assess the run time and memory requirements of the proposed exploratory trajectory inference approach across a larger range of dataset sizes, we resorted to a dataset of mouse gastrulation [25], downloaded from scVelo (version 0.3.3) [14]. To evaluate the relation between the number of cells and computational efficiency, the dataset was subsampled to different sizes (5,000, 10,000, 20,000, and 50,000 cells), and exploratory trajectory inference was run on each subsample dataset with four different settings for the number of chains (100, 500, 1,000, and 2,000 chains, S11 Fig). Due to the distance calculations between simulated chains, our method scales quadratically with the number of chains (S11 Fig D). The effect of the number of cells on run time (S11 Fig C) is linked to the number of steps required to reach the stationary distribution S11 Fig E and F): With increasing size of the dataset, the number of required steps also rises.

References:

[1] Zheng, S.C., Stein-O’Brien, G., Boukas, L. et al. Pumping the brakes on RNA velocity by understanding and interpreting RNA velocity estimates. Genome Biol 24, 246 (2023). https://doi.org/10.1186/s13059-023-03065-x

Reviewer #2: The authors proposed a novel framework termed exploratory trajectory inference to address limitations in resolving complex lineage topologies, particularly convergent differentiation trajectories, from single-cell RNA sequencing (scRNA-seq) data. By applying this approach to an original dataset of CD8 T cell responses in chronic LCMV infection, the authors identified several distinct yet converging developmental paths to T cell exhaustion. The inferred trajectories are experimentally supported through adoptive transfer studies. I find the framework proposed in the paper interesting, and the experimental validation is convincing. However, I think the authors should provide more clarification regarding the advances over their previous studies, as well as include additional simulation benchmarking results to better demonstrate the performance of the approach.

Here are my specific comments:

1. The proposed framework, exploratory trajectory inference, appears to be based on cy2path (Ref. 35), which seems closely related to Cytopath (Ref. 14). Given the strong methodological overlap among these works, could the authors clarify how the current manuscript advances beyond these prior studies? Specifically, it would be helpful to delineate which aspects of the methodology are novel in this manuscript, and how it extends or differs from cy2path and/or Cytopath. For example, the description in the Exploratory Trajectory Inference subsection of the Methods section could more clearly highlight the conceptual differences.

Cytopath implements the initial simulation-based trajectory inference approach as described in Gupta et al. 2022. Cytopath assumes predefined start and end states to carry out this task. Cy2path comprises different extensions of the Cytopath procedure. For this study, we resort to the reimplementation of the trajectory inference procedure in the cy2path package, where we consider Markov chain simulations with predefined start states only, i.e., without specifying valid end states for the simulations considered for further analysis. The methodological novelty of the presented exploratory trajectory inference approach consists of (1) considering such simulations and (2) downstream unsupervised summarization of the ensemble of simulations by hierarchical clustering to achieve a set of differentiation lineages. We now describe these conceptual differences between the approaches in the methods section (lines 535-549):

In summary, the main methodological differences of exploratory trajectory inference compared to the Cytopath approach described in [4], enabling a more unbiased analysis, are as follows. Briefly, (1) end states are not explicitly specified. Instead, a fixed number of simulations is performed, with the number of steps determined through convergence to the stationary state distribution. In contrast, Cytopath aims to sample chains that reach the specified end points. (2) Trajectories are inferred based on these simulations by hierarchical, agglomerative clustering with dynamic time warping distance is used instead of clustering with HDBSCAN using Hausdorff distances. In principle, agglomerative clustering enables the analysis of the cluster hierarchy to obtain trajectories at different granularities, in particular for complex differentiation processes. Dynamic time warping distance was chosen instead of Hausdorff distances because it offers a more intuitive approach to compare cell sequences that are not temporally synchronized. Furthermore, Cytopath also uses dynamic time warping to align sequences before computing Hausdorff distances between them. Thus, we chose the more straightforward approach of directly using dynamic time warping distance for exploratory trajectory inference.

2. RNA velocity methods like scVelo are based on abundance quantification and involve numerous steps and parameters, which can lead to variability in results (https://doi.org/10.1371/journal.pcbi.1008585; https://doi.org/10.1371/journal.pcbi.1010492). This variability can propagate downstream through the state transition matrix, potentially affecting trajectory inference outcomes. Can the authors provide some explanation or further analysis on whether exploratory trajectory inference is sensitive to these factors?

To address the important question of whether our exploratory trajectory inference is sensitive to differences in RNA velocity estimates, we qualitatively compared inferred trajectories across a range of values for the number of neighbors used for the estimation of RNA velocity, as well as for all three velocity modes implemented in scVelo. In total we considered 12 different combinations of these parameters and velocity estimation approaches. The results are shown in S12 Fig, S13 Fig, and S15 Fig and described in the discussion section (lines 343-360). In brief, the inferred trajectories are qualitatively mostly stable across the range of tested parameters. In particular, for the stochastic model of RNA velocity, the inferred trajectories are highly consistent across the different values of the number of neighbors, with the only difference being the failure to identify the transition from memory-like to

---

## [Decision Letter · Decision Letter 1]

1 Jul 2025

PONE-D-25-14715R1Exploratory trajectory inference reveals convergent lineages for CD8 T cells in chronic LCMV infectionPLOS ONE

Dear Dr. Claassen,

Thank you for submitting your manuscript to PLOS ONE. After careful consideration, we feel that it has merit but does not fully meet PLOS ONE’s publication criteria as it currently stands. Therefore, we invite you to submit a revised version of the manuscript that addresses the points raised during the review process.

We look forward to receiving your revised manuscript.

Kind regards,

Xianmin Zhu

Academic Editor

PLOS ONE

**Journal Requirements:**

Reviewers' comments:

Reviewer's Responses to Questions

**Comments to the Author**

1. If the authors have adequately addressed your comments raised in a previous round of review and you feel that this manuscript is now acceptable for publication, you may indicate that here to bypass the “Comments to the Author” section, enter your conflict of interest statement in the “Confidential to Editor” section, and submit your "Accept" recommendation.

Reviewer #1: All comments have been addressed

Reviewer #2: All comments have been addressed

Reviewer #3: (No Response)

2. Is the manuscript technically sound, and do the data support the conclusions?

Reviewer #1: Yes

Reviewer #2: Yes

Reviewer #3: Partly

3. Has the statistical analysis been performed appropriately and rigorously? 

Reviewer #1: Yes

Reviewer #2: Yes

Reviewer #3: Yes

4. Have the authors made all data underlying the findings in their manuscript fully available?

Reviewer #1: Yes

Reviewer #2: Yes

Reviewer #3: Yes

5. Is the manuscript presented in an intelligible fashion and written in standard English?

Reviewer #1: Yes

Reviewer #2: Yes

Reviewer #3: Yes

6. Review Comments to the Author

**Reviewer #1: **(No Response)

**Reviewer #2:** (No Response)

**Reviewer #3: **Thank you to the authors for addressing the comments and for the additional analyses and controls now incorporated into the paper.

There are a few remaining comments regarding some of the more qualitative robustness analyses, and clarifications on the uncertainty analysis and metrics for users to determine if an input is less suitable for trajectory inference.

1. To show stability of root/terminal cell probabilities parameters over a range of nearest neighbors (Figure S12, S13) please plot the results in a more quantitative manner (e.g., pairwise correlations between root/terminal probabilities across hyperparameters, or the distributions of these probabilities over the hyperparameters). Otherwise readers need to interpret the grid of embeddings in a qualitative manner.

2. I believe the term ‘diffuse’ is used to define widely distributed root cell probabilities which may lead to harder to interpret trajectories. However, further (quantitative) discussion would be beneficial to make this property of use to users, to be able to assess their data. For example, is there some property of the transition matrix that demonstrate this diffuseness, or can this be plotted in the width/variance of inferred root cell probabilities?

3. Apologies for the potential confusion regarding the comment asking for uncertainty analysis of the trajectories generated over multiple runs. The intent was to assess the variance in the multiple realizations of trajectories in the system, which should be computed in whatever space the trajectories are created in (and clustered etc), which I believe is the 50d PCA space. Assessing variance in the UMAP coordinates adds the layer of a non-deterministic, nonlinear transformation and thus analysis of variance/noise also induced by that process not just through the variance in the trajectories themselves.

Thus this analysis should be performed in the original trajectory space (and be easily available for users to check).

It would also help to visualize variance in these paths in Figure S14, particularly if in scenarios where paths are not recovered correctly and we see more variance/uncertainty in the produced trajectories, i.e., if there is anything a user could use to tell how much to trust or choose between those outputs.

4. For the simulations of the different topologies, please provide details on how many cells/observations were sampled from this model. This Gaussian model also does not represent sparse single-cell count data so the authors should comment on these assumptions (how this may influence results) and demonstrate that the generated points provided to scVelo resemble at least the magnitude of values usually provided in single-cell data.

Additionally, why was a count-based simulator, like dyngen https://github.com/dynverse/dyngen for example, not used here to generate different topologies that also provide ‘realistic’ RNA counts (the usual input to scVelo)?

7. PLOS authors have the option to publish the peer review history of their article (what does this mean?). If published, this will include your full peer review and any attached files.

Reviewer #1: No

Reviewer #2: No

Reviewer #3: No

---

## [Author Response · Author response to Decision Letter 2]

15 Aug 2025

Please find below the point-by-point response to the reviewer comments. We also provide a formated pdf version of these respsonses. Reviewer comments are copied in green font, author responses in black font, and text citations from the revised manuscript are displayed in black italic font. Reference numbers within the cited text refer to the manuscript references.

1. To show stability of root/terminal cell probabilities parameters over a range of nearest neighbors (Figure S12, S13) please plot the results in a more quantitative manner (e.g., pairwise correlations between root/terminal probabilities across hyperparameters, or the distributions of these probabilities over the hyperparameters). Otherwise readers need to interpret the grid of embeddings in a qualitative manner.

We acknowledge that our previous comparison of root cell and endpoint probabilities across hyperparameters used in RNA velocity inference relied on a qualitative interpretation of the grid of embeddings. According to the reviewer's suggestion, we have added a quantitative evaluation in the form of pairwise Spearman’s correlations between probability assignments (S15 Fig). We find these are very similar across velocity estimation procedures and numbers of neighbors (median correlation of 0.99 and 0.91 for root cell and end point probabilities, respectively). We have adjusted the respective description in the Discussion section (lines 348-354):

For the three different models in scVelo (“deterministic”, “stochastic”, and “dynamical”) and four different settings for the number of neighbors (15, 30, 50, and 100), the inferred velocities, root cells (S13 Fig), and end points (S14 Fig) were very similar, as confirmed by high pairwise Spearman’s correlation (S15 Fig), and mostly differed in the position of the root cells within the early cluster and in the magnitude of end cell probabilities of the terminally exhausted and memory-like clusters.

2. I believe the term ‘diffuse’ is used to define widely distributed root cell probabilities which may lead to harder to interpret trajectories. However, further (quantitative) discussion would be beneficial to make this property of use to users, to be able to assess their data. For example, is there some property of the transition matrix that demonstrate this diffuseness, or can this be plotted in the width/variance of inferred root cell probabilities?

We thank the reviewer for the question. To clarify this point, we have added a supplementary figure (S22 Fig) to the manuscript, alongside the following explanation in the Discussion section (lines 420-432):

To judge the diffuseness of terminal states, we suggest assessing the joint and marginal distributions of root cell and end point probabilities (S22 Fig). In a situation of low diffuseness of terminal states, the majority of cells has a root cell and end point probability of zero, few cells have a high probability for either state, the marginal distributions have a low median absolute deviation (MAD), and, most importantly, there are no cells with non-negligible probability for both terminal states. The T cell exhaustion dataset analyzed in this study exhibits the above properties indicative of a situation of low diffuseness of terminal states (S22 Fig A). For the PBMC dataset, we find a situation of high diffuseness, i.e., the majority of cells could, to a varying extent, be either a starting or an end state, and the marginal distributions have a higher MAD

(S22 Fig B). The simulated dyngen cycle dataset lies between these two extremes, with well-defined root cells, but more widely spread end point probabilities (S22 Fig C).

3. Apologies for the potential confusion regarding the comment asking for uncertainty analysis of the trajectories generated over multiple runs. The intent was to assess the variance in the multiple realizations of trajectories in the system, which should be computed in whatever space the trajectories are created in (and clustered etc), which I believe is the 50d PCA space. Assessing variance in the UMAP coordinates adds the layer of a non-deterministic, nonlinear transformation and thus analysis of variance/noise also induced by that process not just through the variance in the trajectories themselves.

Thus this analysis should be performed in the original trajectory space (and be easily available for users to check).

It would also help to visualize variance in these paths in Figure S14, particularly if in scenarios where paths are not recovered correctly and we see more variance/uncertainty in the produced trajectories, i.e., if there is anything a user could use to tell how much to trust or choose between those outputs.

According to the reviewer’s suggestion, we have added an assessment of the variance in inferred trajectories over multiple runs in the 50-dimensional PCA space. For an easily interpretable visualization, we represent this variance for each step of the trajectory by the color of the line showing the mean inferred trajectory coordinate across multiple runs in the UMAP embedding (S10 Fig). We performed the same analysis for the experiments with varying RNA velocity estimation parameters (S16 Fig, formerly S14 Fig), where previously the results of a single execution of exploratory trajectory inference were shown.

The Materials and Methods subsection “Uncertainty estimation of inferred trajectories” was updated to reflect this change (lines 588-592):

The mean and standard deviation of trajectory coordinates in UMAP space, as well as the variance of coordinates in PCA space across the repeated mapped runs, were used to visualize the uncertainty of inferred lineages on the UMAP embedding. For the latter, we visualized the sum of the variance of all 50 principal components.

The results for different numbers of lineages are reported in the Results section, lines 218-220:

We found increasing uncertainty and overlap between lineages with an increasing number of lineages. The same trend was observed in PCA space in the form of the variance for each step across 10 runs, summed over all 50 principal components (S10 Fig).

For the results with varying velocity estimation parameters, the Discussion section was adapted as follows (lines 356-364):

Notably, the only qualitative difference between the trajectories inferred from stochastic velocities across four different neighborhood sizes was the failure to detect the transition from memory-like to terminally exhausted cells with 15 neighbors. The dynamical model resulted in larger deviations for 15 and 30 neighbors, but produced similar qualitative trajectories for 100 neighbors. However, for the two strongly deviating results based on the dynamical model, the inferred trajectories also showed a higher variance across repeated runs than those for the stochastic model. Our findings reported in this study, obtained with the stochastic model and 30 nearest neighbors, were supported by the majority of trajectory inference analyses with the other considered parameters.

4. For the simulations of the different topologies, please provide details on how many cells/observations were sampled from this model. This Gaussian model also does not represent sparse single-cell count data so the authors should comment on these assumptions (how this may influence results) and demonstrate that the generated points provided to scVelo resemble at least the magnitude of values usually provided in single-cell data.

Additionally, why was a count-based simulator, like dyngen https://github.com/dynverse/dyngen for example, not used here to generate different topologies that also provide ‘realistic’ RNA counts (the usual input to scVelo)?

We have expanded our description of the data generation in the Materials and Methods section to indicate the number of generated observations (lines 636-642):

To generate positive control data with simple, well-known ground truth trajectories of varying topology, we constructed trajectories in two dimensions (see left column in S19 Fig). To this end, we defined cluster means forming the desired topology and sampled 100 points from a bivariate Gaussian distribution around the means. Additionally, we sampled interpolated points lying between the main clusters (100 points for each pair of consecutive clusters). This resulted in 1,300, 3,700, 2,000, and 2,400 data points for the bifurcating, double bifurcating, converging, and cyclical topologies, respectively.

We acknowledge that these simulations represent a strong simplification and do not represent the statistical properties of single-cell RNA sequencing data. According to the reviewer’s comment, we have now included dyngen to generate more realistic simulated data, as discussed in lines 387-405.

Specifically, we simulated data from processes constructed in 2D (see Simulation of positive control data for details on data generation). [...] While showcasing the capability of our method to infer lineages in situations with velocities strongly aligned with the defined dynamic process, it does not represent the statistical properties of sparse scRNAseq count data. Therefore, to generate more realistic simulations with known trajectories, we additionally used dyngen [29], with various backbones, resulting in diverse process topologies. Here, exploratory trajectory inference showed overall good performance in recovering the true trajectories as well (S20 Fig). Automatic selection yielded the correct number of lineages for all backbones except for the cycle. Here, we obtained three lineages instead of one, all following the cyclic dynamic process. However, this simulated dataset was particularly challenging since the inferred end points are distributed all along the cyclic path, and the ground truth simulation also showed multiple cycles (S20 Fig D-F), in contrast to, e.g., the bifurcating backbone, where end points were more restricted and where there were no cyclical patterns (S20 Fig A-C).

The data generation with dyngen is described in detail in the Materials and Methods section, lines 651-666:

Additional positive control datasets were generated with dyngen [29] in R using the bifurcating, cycle, consecutive bifurcating, trifurcating, converging, and disconnected backbones. For each dataset, 500 genes were simulated for 5,000 cells. The number of transcription factors was set to the number of modules in the respective backbone. Half of the remaining genes were simulated as targets, the rest as housekeeping genes. The simulation parameters were set as follows: total_time = 1000, census_interval = 2, ssa_algorithm = ssa_etl(tau = 300 / 3600). We performed wild-type simulations with 500 simulations for a snapshot experiment, computing RNA velocity as well. The data was further processed using Scanpy and scVelo, using the simulated counts, spliced counts, unspliced counts, and RNA velocity. Genes with less than 20 shared counts (spliced and unspliced) were removed, and the three count matrices were normalized to the respective median of total counts. The expression data was log-transformed and scaled. PCA, a neighborhood graph, and a UMAP projection were computed with default parameters. Normalized spliced and unspliced counts were smoothed using scvelo.pp.moments, and a velocity graph was built using neighbor connectivities. Terminal states and a transition matrix were computed.

---

## [Decision Letter · Decision Letter 2]

31 Aug 2025

Exploratory trajectory inference reveals convergent lineages for CD8 T cells in chronic LCMV infection

PONE-D-25-14715R2

Dear Dr. Claassen,

We’re pleased to inform you that your manuscript has been judged scientifically suitable for publication and will be formally accepted for publication once it meets all outstanding technical requirements.

Kind regards,

Xianmin Zhu

Academic Editor

PLOS ONE

Additional Editor Comments (optional):

Reviewer #3:

Reviewers' comments:

Reviewer's Responses to Questions

**Comments to the Author**

1. If the authors have adequately addressed your comments raised in a previous round of review and you feel that this manuscript is now acceptable for publication, you may indicate that here to bypass the “Comments to the Author” section, enter your conflict of interest statement in the “Confidential to Editor” section, and submit your "Accept" recommendation.

Reviewer #3: All comments have been addressed

2. Is the manuscript technically sound, and do the data support the conclusions?

Reviewer #3: Yes

3. Has the statistical analysis been performed appropriately and rigorously? 

Reviewer #3: Yes

4. Have the authors made all data underlying the findings in their manuscript fully available?

Reviewer #3: Yes

5. Is the manuscript presented in an intelligible fashion and written in standard English?

Reviewer #3: Yes

6. Review Comments to the Author

Reviewer #3: The authors have sufficiently addressed the reviewer comments and provided thorough evaluations of the outputs of their method with quantitative metrics for user interpretation of inference results.

7. PLOS authors have the option to publish the peer review history of their article (what does this mean?). If published, this will include your full peer review and any attached files.

Reviewer #3: No

---

## [Editor Report · Acceptance letter]

PONE-D-25-14715R2

PLOS ONE

Dear Dr. Claassen,

I'm pleased to inform you that your manuscript has been deemed suitable for publication in PLOS ONE. Congratulations! Your manuscript is now being handed over to our production team.

Kind regards,

on behalf of

Dr. Xianmin Zhu

Academic Editor

PLOS ONE